# Adenosine and acute low oxygen conditions suppress urinary bladder contractility through the activation of adenosine 2B receptors and large-conductance calcium-activated potassium channels

Gerald M. Herrera[1] (iD), Jason L. Rengo[1,2] (iD), Grant W. Hennig[1] (iD), Thomas J. Heppner[1] (iD), Alexandria M. Hepp[1], Maria Sancho[1,3] (iD), Saul Huerta de la Cruz[1] (iD), Mark T. Nelson[1,4] (iD) and Nicholas R. Klug[1] (iD)

[1] *Department of Pharmacology, Larner College of Medicine, University of Vermont, Burlington, Vermont, USA*
[2] *Cellular, Molecular, and Biomedical Sciences Graduate Program, University of Vermont, Burlington, Vermont, USA*
[3] *Department of Physiology, Faculty of Medicine, Complutense University of Madrid, Madrid, Spain*
[4] *Division of Cardiovascular Sciences, University of Manchester, Manchester, UK*

Handling Editors: Peying Fong & Bernard Drumm

The peer review history is available in the Supporting Information section of this article (https://doi.org/10.1113/JP289080#support-information-section).

**Abstract figure legend** Detrusor smooth muscle surrounding the urinary bladder produces transient pressure events essential for sensing bladder fullness during the filling phase. We investigated the role of adenosine, a ubiquitous signalling molecule, in regulating bladder smooth muscle function. Adenosine activates A2B receptors and downstream large-conductance calcium-activated potassium (BK$_{Ca}$) channels, leading to the suppression of smooth muscle-driven transient pressure events. Additionally acute hypoxia, known to engage adenosine signalling in other tissues, relaxes bladder smooth muscle via A2B receptor activation. These findings underscore the inhibitory influence of adenosine signalling and acute hypoxia on bladder contractility.

This article was first published as a preprint. Herrera GM, Heppner TJ, Hennig GW, Rengo JL, Hepp AM, Sancho M, Huerta de la Cruz S, Nelson MT, Klug NR. 2025. Adenosine and acute low oxygen conditions suppress urinary bladder contractility through the activation of adenosine 2B receptors and large-conductance calcium-activated potassium channels. bioRxiv. https://doi.org/10.1101/2025.04.14.648772

The Journal of Physiology

**Abstract** Under healthy conditions the urinary bladder undergoes relatively long periods of filling with well-spaced voiding events to ensure proper storage and removal of urine. During the filling phase distinct contractile events in the urinary bladder smooth muscle (UBSM) comprising the detrusor elicit transient non-voiding pressure events and associated bursts in afferent nerve activity to relay the sensation of bladder fullness. The mechanisms that regulate UBSM excitability and associated non-voiding pressure events under physiological and pathological conditions are poorly understood. Here we investigated the role of adenosine signalling in regulating urinary bladder contractility. Using an *ex vivo* pressurized bladder preparation from mice and patch-clamp electrophysiology in isolated UBSM cells, we evaluated whole bladder transient pressure events, whole bladder detrusor $Ca^{2+}$ activity and single UBSM ion channel activity. We found that adenosine suppresses bladder activity through the activation of A2B adenosine receptors and down-stream activation of large-conductance calcium-activated potassium ($BK_{Ca}$) channels. We further demonstrated that acute exposure to low oxygen conditions using a chemical oxygen scavenger potently suppresses bladder contractility through the A2B receptor pathway. These results highlight the prominent role adenosine receptors and downstream potassium channels play in regulating urinary bladder contractility in physiological and pathological contexts.

(Received 16 April 2025; accepted after revision 28 September 2025; first published online 4 November 2025)

**Corresponding authors** N. R. Klug and G. M. Herrera: Department of Pharmacology, Larner College of Medicine, University of Vermont, 149 Beaumont Avenue FMRB 454, Burlington, VT 05405, USA. Email: nicholas.klug@uvm.edu and gerald.herrera@uvm.edu

**Key points**

- This study shows that adenosine, a signalling molecule, reduces bladder contractility by activating A2B receptors and large-conductance calcium-activated potassium ($BK_{Ca}$) channels.
- Low oxygen conditions also suppress bladder activity through the activation of A2B receptors, linking acute hypoxia to bladder relaxation.
- ATP-sensitive potassium ($K_{ATP}$) channels, often involved in muscle relaxation, do not contribute to adenosine's effects in the bladder.
- These findings reveal a new pathway that controls bladder function and may help explain bladder disorders related to low oxygen, such as overactive or underactive bladder.

## Introduction

Under healthy conditions the urinary bladder painlessly stores and voids urine at intervals that are not disruptive to daily living. During the filling or storage phase, distinct cell types within the bladder, such as the inner urothelial cell layer, the outer detrusor smooth muscle layer and the embedded sensory nerves, communicate to relay the sensation of fullness (Birder & Andersson, 2013; Grundy et al., 2024; Merrill et al., 2016). Once full voiding occurs through parasympathetic stimulation of the detrusor smooth muscle layer and relaxation of the internal urethral sphincter, and voluntary relaxation of the external urethral sphincter (Andersson & Arner, 2004). Although the urothelium and detrusor have distinct functions such as establishing a tight barrier and providing

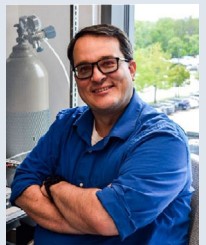

Gerald M. Herrera is a member of the pharmacology faculty at the Larner College of Medicine at the University of Vermont. His work investigates mechanisms regulating lower urinary tract function. He makes use of multifaceted techniques – from single-cell electrophysiology and calcium imaging to *in vivo* models – to delineate how the bladder senses fullness. He has made important contributions to understanding excitation–contraction coupling in urinary bladder smooth muscle. Prior to returning to academia in 2019, he directed research and development at a leading biomedical instrumentation company. His diverse experience and skillset inform his integrative approach to science and mentorship, bridging basic science research with real-world applications.

voiding pressure, respectively, signalling between these layers is critical for healthy bladder function. Indeed disruptions to key paracrine signalling processes between the urothelium and detrusor can result in a number of lower urinary pathologies causing profound detriments to quality of life, such as overactive and underactive bladder, bladder ischaemia, urge incontinence and bladder pain (Andersson & Arner, 2004; Khandelwal et al., 2009; Merrill et al., 2016). Although the urinary bladder smooth muscle (UBSM) is typically described as 'relaxed' during the bladder filling phase, it exhibits considerable dynamic activity ranging from individual fibre action potentials to short-propagating contractile waves. This activity produces phasic pressure events in the bladder, which regulate afferent nerve activity and contribute to the sensation of fullness (Heppner et al., 2016). Thus understanding the mechanisms that regulate normal and dysfunctional UBSM activity during filling, as well as the intrinsic signalling mechanisms between distinct bladder layers, may provide key insights into bladder function and pathology.

Among the key regulators of bladder function, ATP released by the efferent bladder nerves and/or the urothelium onto UBSM has an excitatory effect by activating P2Y and P2X purinergic receptors (Andersson, 2015; Ruggieri, 2006). However once released extracellular ATP is rapidly degraded by ectonucleotidase enzymes into adenosine, which activates G protein-coupled receptors (GPCR), that is adenosine A1, A2 and A3 receptors (Eltzschig et al., 2003; Kleppisch & Nelson, 1995; Mubagwa & Flameng, 2001; Sancho et al., 2022; Sebastiao & Ribeiro, 2009). Adenosine signalling mechanisms play a critical role in modulating the excitability of neurons, striated muscle, endothelial cells, vascular smooth muscle and pericytes (Burnstock, 2017; Kleppisch & Nelson, 1995; Sancho et al., 2022; Sebastiao & Ribeiro, 2009). In the bladder adenosine exerts a relaxing or inhibitory effect on detrusor contractility by activating A2B receptors and not A2A receptors (Hao et al., 2019; Pakzad et al., 2016). Additionally adenosine signalling is critical during hypoxic or ischaemic conditions, where adenosine levels increase in response to metabolic stress. This elevation in adenosine is a key factor in the progression of ischaemic diseases (Fredholm et al., 2005; Karmouty-Quintana et al., 2013; Zheng et al., 2007). However the downstream mechanisms by which adenosine modulates urinary bladder activity, and its specific role during bladder ischaemia/hypoxia, remain unclear.

Here using *ex vivo* urinary bladder preparations and single isolated UBSM cells from mice, we examined the effect of adenosine on bladder contractile activity and the downstream signalling mechanisms that regulate UBSM contractility. We found that adenosine, via A2B receptor activation, suppresses UBSM-mediated trans-ient pressure events upstream of large-conductance calcium-activated potassium ($BK_{Ca}$) channels but not ATP-sensitive potassium ($K_{ATP}$) channels. Further we investigated the link between hypoxia, adenosine signalling and bladder contractility within the urinary bladder. We found that chemical oxygen depletion significantly relaxes UBSM and prevents phasic pressure events through intrinsic activation of A2B receptors.

## Results

### Urinary bladder contractility is reduced by adenosine in an A2B receptor-cAMP-dependent manner

To examine contractility from an intact whole urinary bladder, dissected bladders from mice were placed in a perfusion chamber and cannulated. The cannulation line was connected to a syringe pump to fill the bladder, and an inline pressure transducer recorded transient pressure events resulting from phasic contractile events produced by UBSM (Heppner et al., 2016). Transient pressure event amplitude and frequency for each condition were analysed during a 5-min isovolumetric period after an intravesical infusion of the bladder to 12 mmHg (Fig. 1A). Bladders were isolated from either myh11-CCaMP6f mice on a C57bl/6J background that express inducible genetically encoded calcium indicator, GCaMP6f, in smooth muscle, or wild-type C57bl/6J mice.

The baseline transient pressure event amplitude was $0.89 \pm 0.71$ mmHg, and the frequency was $4.92 \pm 1.32$ events/min. Bath superfusion of the metabolically stable adenosine analogue 2-chloroadenosine (CADO, 5 µM) significantly reduced transient pressure event amplitude and frequency by 90% and 85%, respectively (Fig. 1B). This relaxing effect was eliminated by pre-incubation with A2B receptor antagonist PSB 603 (500 nM, Fig. 1B), suggesting that the relaxing effects of adenosine occur through A2B receptor-mediated Gs protein-coupled receptor (GsPCR) stimulation and down-stream cAMP to protein kinase A (PKA) signalling. Indeed forskolin (5 µM), an adenylate cyclase activator, mimicked the relaxing effects of CADO even in presence of A2B receptor blockade (Fig. 1B), confirming that the mechanism is cAMP dependent. Representative traces of transient pressure events for the different experimental conditions are shown in Fig. 1C.

Recording pressure events from the whole bladder provides a surrogate for the total force exerted by UBSM contractile activity but offers no insight into the spatial temporal patterning and co-ordination of this activity. Using an unbiased standard deviation method to extract UBSM $Ca^{2+}$ events (see Movie 1), the overall amount of $Ca^{2+}$ activity at every point on the bladder surface was quantified as prevalence (seconds active per minute, s/min; Fig. 2A). Local co-ordination of $Ca^{2+}$ activity

between UBSM bundles was used to examine regional differences in how activity spread throughout the bladder, quantified as coincidence (% neighbourhood search area containing synchronous activity, %Max; Fig. 2*A*). The temporal dynamics of these parameters were plotted as spatio-temporal maps (Fig. 2*A*) and were further condensed into single values representing the overall amount (prevalence) and synchrony (coincidence) of activity throughout the entire hemispherical bladder surface during each recording (Fig. 2*B* and *C*).

In response to CADO (5 µM) prevalence and coincidence measures of $Ca^{2+}$ activity were reduced to 20.7% and 31.8% of baseline values, respectively (Fig. 2*B* and *C*), indicating both the amount of $Ca^{2+}$ activity and its propagation were significantly affected. Preincubation of PSB 603 (500 nM) had no effect on baseline prevalence or coincidence and completely prevented the inhibitory effects of CADO (Fig. 2*B* and *C*). Forskolin (5 µM) potently inhibited prevalence and coincidence measures of UBSM $Ca^{2+}$ events in presence of PSB 603 (Fig. 2*B* and *C*). Movie 2 demonstrates the effect of CADO, and Movie

3 demonstrates the effects of PSB 603, PSB 603 + CADO and PSB 603 + forskolin on urinary bladder $Ca^{2+}$ activity.

Overall these results suggest that adenosine-induced bladder relaxation is mediated via A2B receptors through a cAMP-dependent mechanism. To further dissect the downstream pathways involved in this relaxation, we next explored the role of prominent cAMP-PKA-sensitive potassium channels expressed in UBSM.

## Adenosine-induced bladder relaxation is mediated by increased $BK_{CA}$ channel activity but has no effect on $K_{ATP}$ channels

Adenosine signalling via A2B receptor activation promotes downstream cAMP production and PKA activation. In many types of smooth muscle, PKA signalling exerts potent relaxing effects through multiple mechanisms, including ion channel modulation and inhibition of cross-bridge formation (Kleppisch & Nelson, 1995; Murthy, 2006; Porter et al., 1998; Wellman & Nelson,

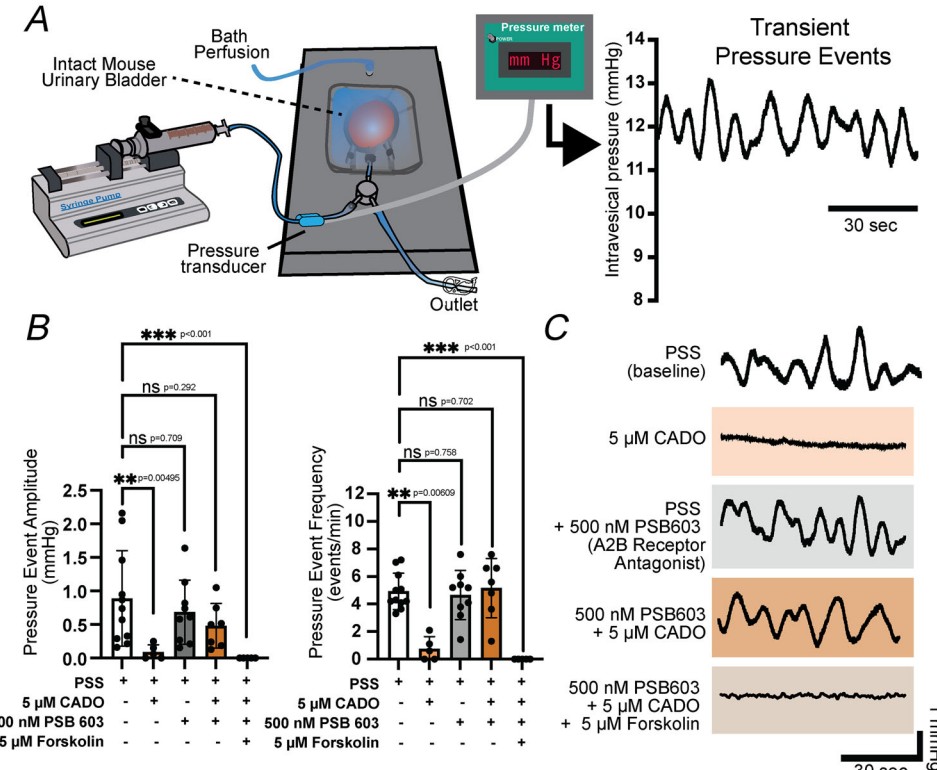

**Figure 1. Adenosine reduces whole bladder transient pressure event amplitude and frequency in an A2B receptor-dependent manner**

*A*, illustration of whole bladder cannulation and perfusion set-up, where whole mouse bladder is pressurized and bath perfused. Phasic contractile activity is observed as transient pressure events using an inline pressure transducer. *B*, summarized effects of physiological salt solution (PSS, baseline condition), 2-chloroadenosine (CADO, 5 µM), A2B receptor antagonist PSB 603 (500 nM) and adenylate cyclase activator forskolin (5 µM). Data generated during a 5-min interval after at least 15 min of drug/compound exposure. *C*, representative pressure traces during each respective condition from summary data in (*B*). Data shown are mean ± SD; *N* = 5–9 per group; individual *P*-values shown using Kruskal–Wallis test unless ***$P$ < 0.001.

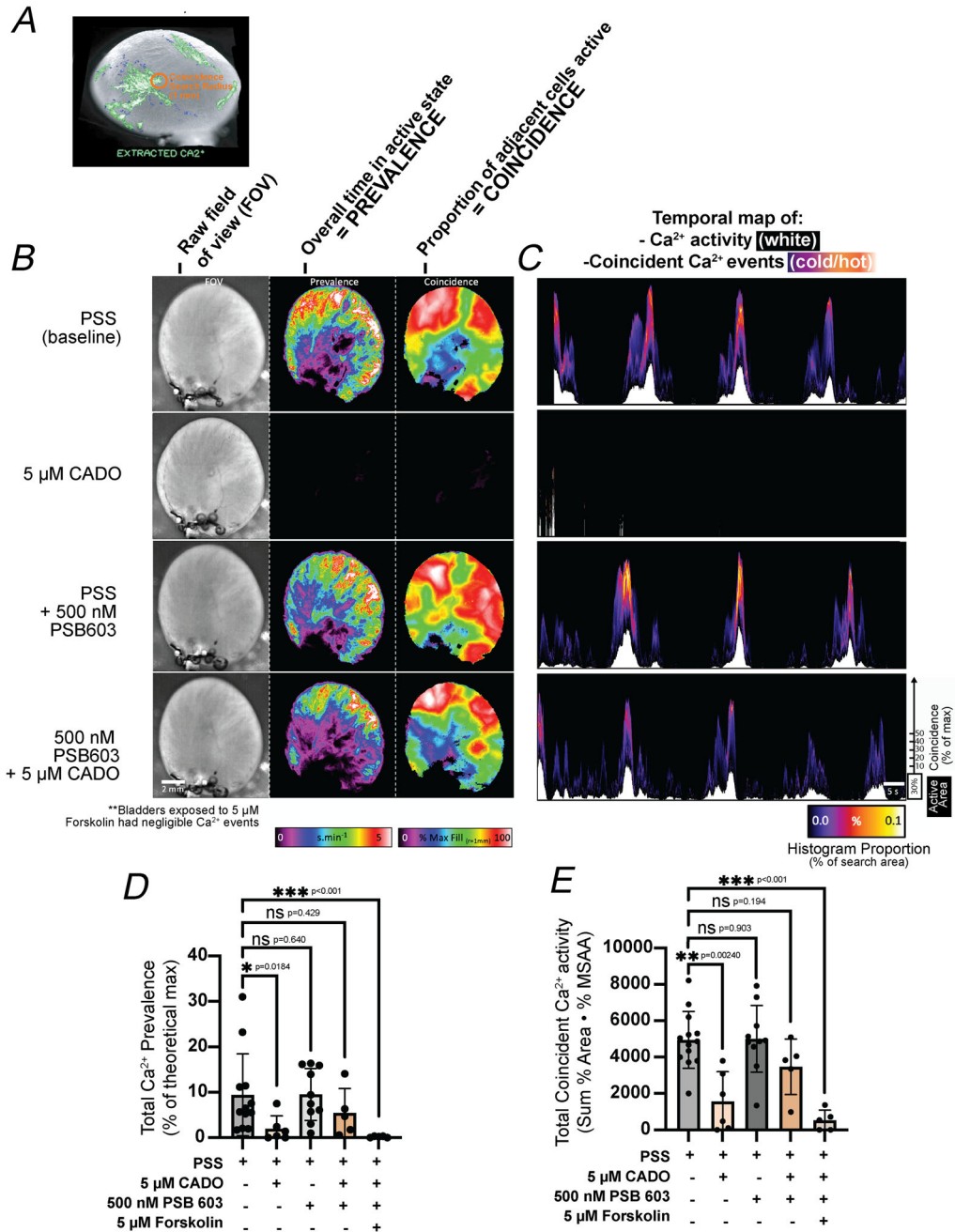

**Figure 2. Adenosine reduces urinary bladder smooth muscle (UBSM) calcium activity in an A2B receptor-dependent manner**

*A*, illustration of whole bladder $Ca^{2+}$ imaging using a macroview widefield microscope. Extracted $Ca^{2+}$ events used to quantify overall prevalence and coincidence patterning of $Ca^{2+}$ activity from UBSM layer. *B*, representative experiment showing FOV (field of view) (left) and parameter overlays of prevalence (cumulative $Ca^{2+}$ activity, middle) and maximum coincidence (maximum proportion of simultaneous firing in a small search window, radius = 1 mm) calculated at every point on the bladder surface over the entire recording duration. *C*, spatio-temporal maps of combined prevalence (white trace, bottom) and coincidence (proportion of coincidence values mapped as cold/hot, with the actual coincidence value from 0% to 100% plotted on the *y*-axis superimposed on top of the prevalence trace). *D*, summarized effects of physiological salt solution (PSS, baseline condition), 2-chloroadenosine (CADO, 5 µM), A2B receptor antagonist PSB 603 (500 nM) and adenylate cyclase activator forskolin (5 µM) on prevalence. *E*, summarized effects of conditions from (*D*) on maximum coincidence. Data shown are mean ± SD; *N* = 5–9 per group; individual *P*-values shown using Kruskal–Wallis test unless ***P* < 0.001.

2003). Figure 3*A* shows the potential downstream targets of adenosine signalling in UBSM.

Among these targets $K_{ATP}$ channels, consisting of Kir 6.1 and SUR2 subunits, are expressed in UBSM (Brayden, 2002; Malysz & Petkov, 2020; Petkov et al., 2001) and are a canonical downstream target of the GsPCR-cAMP-PKA signalling cascade (Quinn et al., 2004; Shi et al., 2008). Phosphorylation of $K_{ATP}$ channels by PKA enhances their activity, leading to $K^+$ efflux and hyperpolarization of the smooth muscle membrane, a mechanism that induces a potent relaxing effect (Brayden, 2002; Petkov et al., 2001). Thus we hypothesized that activation of $K_{ATP}$ channels may underlie adenosine-induced relaxation of UBSM.

To test this $K_{ATP}$ channels were pharmacologically activated using the synthetic activator pinacidil (10 μM) in *ex vivo* mouse bladder preparations (C57bl/6J mice). Pinacidil application significantly reduced transient pressure event amplitude and frequency by 84.8% and 86.6%, respectively (Fig. 3*B* and *C*). Blocking $K_{ATP}$ channels with glibenclamide (10 μM) had no effect on baseline transient bladder amplitude or frequency, indicating that $K_{ATP}$ channels do not significantly contribute to baseline bladder activity/excitability. Furthermore glibenclamide prevented the relaxing effects of pinacidil (Fig. 3*B* and *C*), indicating that UBSM express functional $K_{ATP}$ channels.

Next we investigated whether $K_{ATP}$ channels are involved in the relaxing effects of A2B receptor activation. Preincubation of *ex vivo* bladders with glibenclamide and subsequent application of CADO did not significantly alter the relaxing effect of CADO, with a 93.7% and 91.4% reduction in transient pressure event amplitude and frequency, respectively (Fig. 3*D* and *E*). These findings suggest that $K_{ATP}$ channels are not engaged downstream of A2B receptor activation in UBSM.

Another potential downstream target of adenosine signalling is the $BK_{Ca}$ channel, which is functionally present in UBSM and plays a critical repolarization role in the UBSM action potential (Heppner et al., 1997). $BK_{Ca}$ channel activity is also enhanced by the GsPCR-cAMP-PKA signalling cascade through direct channel phosphorylation and increased $Ca^{2+}$ release via ryanodine receptor activity (Sancho & Kyle, 2021; Schubert & Nelson, 2001). To test the involvement of $BK_{Ca}$ channels in adenosine-induced relaxation, we preincubated the bladder with the $BK_{Ca}$ channel blocker paxilline (5 μM). Paxilline enhanced the transient pressure event amplitude and frequency by 116.6% and 111.3%, respectively (Fig. 3*F* and *G*). Subsequent exposure to 5 μM CADO did not significantly reduce pressure event amplitude or frequency (Fig. 3*F* and *G*). Overall these results indicate the following: (1) unlike $K_{ATP}$ channels in other types of smooth muscle, UBSM $K_{ATP}$ channels do not contribute to UBSM relaxation in response to adenosine signalling; (2) during $BK_{Ca}$ channel inhibition adenosine has no effect on UBSM contractility, suggesting a critical role for $BK_{Ca}$ channels downstream of adenosine receptors; and (3) PKA effects on myofilament force production is not the predominant mechanism for adenosine-induced bladder relaxation because $K^+$ channel inhibition, alone, elicited prominent phasic contractions that were unaffected by adenosine administration.

### A2B receptor stimulation enhances $BK_{Ca}$ channel activity in isolated UBSM cells

Figure 3*F* and *G* shows that inhibiting $BK_{Ca}$ channels in whole bladder significantly alters baseline contractility. Therefore we further assessed the direct effects of adenosine signalling on $BK_{Ca}$ channel activity using freshly isolated UBSM and patch-clamp electrophysiology (Fig. 4*A*). Using whole-cell configuration and 300 ms voltage steps from –70 to +70 mV (Fig. 3*B*), UBSM exhibited prominent outward currents that were sensitive to paxilline (1 μM), a specific $BK_{Ca}$ channel blocker (Fig. 4*B* and *C*). Baseline paxilline-sensitive outward currents at +70 mV were 20.1 ± 2.2 pA/pF. Maximal outward currents (at +70 mV) were enhanced by 75% after the addition of 5 μM CADO to the bath solution (Fig. 4*B* and *C*). To determine whether this enhancement was mediated by A2B receptors, UBSM cells were preincubated with the A2B receptor antagonist PSB 603 (500 nM). In presence of PSB 603 the adenosine-induced enhancement of outward currents was significantly diminished (Fig. 4*D–F*). These results suggest that adenosine enhances $BK_{Ca}$ channel activity in UBSM through an A2B receptor-dependent pathway.

### Urinary bladder contractility is reduced by low oxygen conditions in an A2B receptor-dependent manner

Because adenosine signalling is critical to regulate hypoxic and ischaemic responses in other contractile tissues and organs (Fredholm et al., 2005; Karmouty-Quintana et al., 2013; Zheng et al., 2007), we investigated whether acute low oxygen conditions alter whole bladder transient pressure events (i.e. whole bladder contractility) in an adenosine receptor-dependent manner. Addition of 10 mM of the oxygen scavenger sodium sulphite ($Na_2SO_3$) and bubbling physiological salt solution (PSS) with 95% $N_2$ and 5% $CO_2$ induced severe hypoxic conditions (Jiang et al., 2011; Marino et al., 2020). Under these conditions the average bath oxygen percentage was measured at 0.1%–0.2% (Fig. 5*A*). Because bath oxygen percentage was not measured at 0% $O_2$, we classified these bath conditions as severely hypoxic. Exposure to hypoxic conditions reduced transient pressure events amplitude

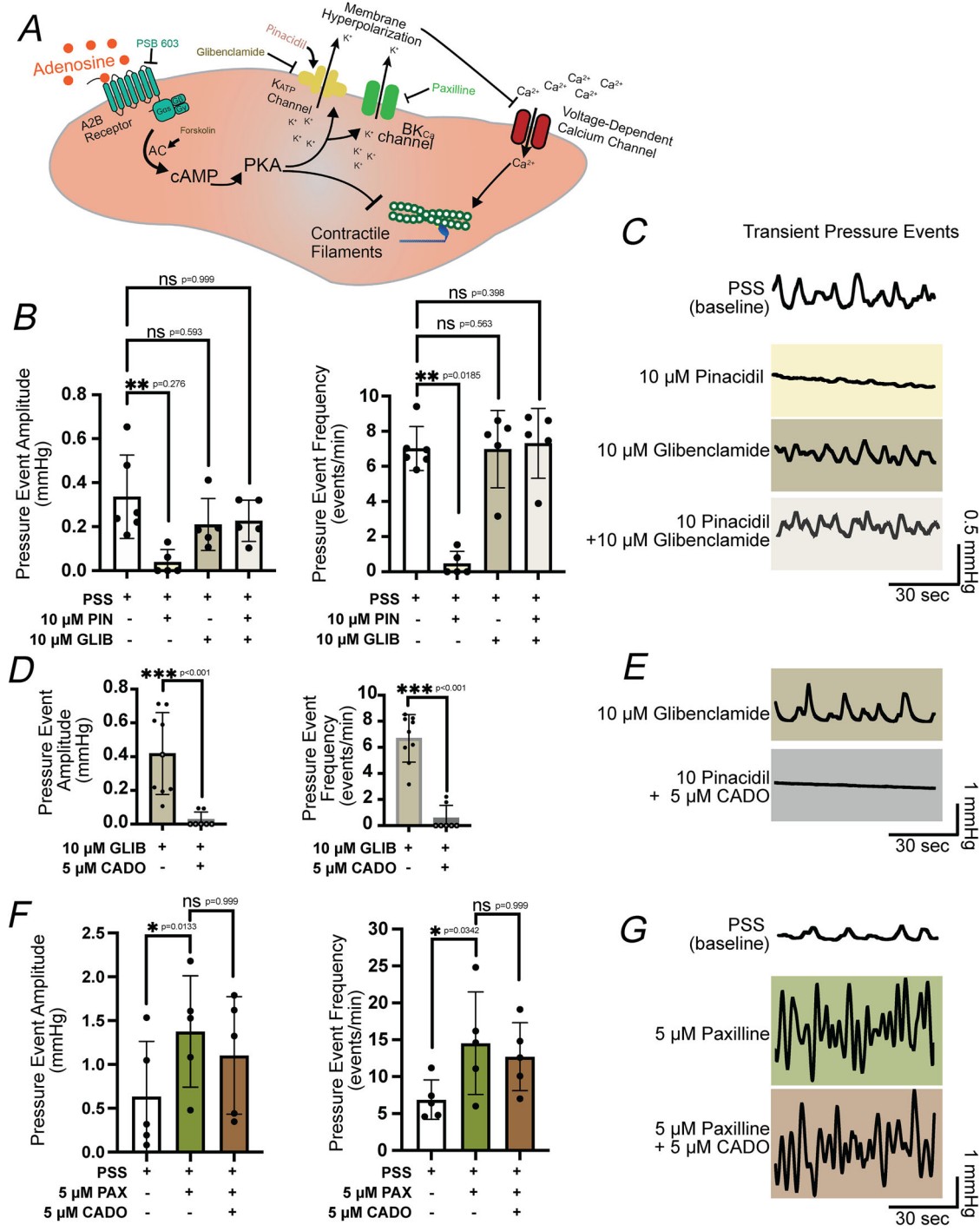

**Figure 3. Adenosine-induced bladder relaxation does not rely on ATP-sensitive potassium (K$_{ATP}$) channel activation and is reversed by inhibition of large-conductance calcium-activated potassium channels**
*A*, schematic illustrating potential downstream targets of A2B receptor activation that would reduce bladder contractility. Logical downstream targets include hyperpolarization by protein kinase A (PKA)-sensitive potassium channels, K$_{ATP}$ and large-conductance calcium-activated potassium (BK$_{Ca}$) channels, and reduced contractility through PKA-dependent inhibition of myofilament activity. *B*, summarized effects of physiological salt solution (PSS, baseline condition), K$_{ATP}$ channel activation by pinacidil (PIN, 10 µM) and K$_{ATP}$ channel inhibition by glibenclamide (GLIB, 10 µM). *C*, representative pressure traces during each respective condition from summary data in (*B*). *D*, summarized effects of glibenclamide treatment (GLIB, 10 µM) and 2-chloroadenosine (CADO, 5 µM). *E*, representative pressure traces during each respective condition from (*D*). *F*, summarized effects of physiological salt solution (PSS, baseline condition), BK$_{Ca}$ channel inhibition by paxilline (PAX, 5 µM) and CADO (5 µM). *G*,

representative pressure traces during each respective condition from (*F*). Data generated during a 5-min interval after at least 15 min of drug/compound exposure. Data shown are mean ± SD; *N* = 5–10 per group; individual *P*-values shown using Kruskal–Wallis or Friedman test (for *F*, paired values) test unless \*\*\**P* < 0.001.

and frequency by 90.9% and 69.1%, respectively (Fig. 5*B* and *C*), indicating a profound suppression of bladder contractility. To determine whether this suppression was mediated by adenosine signalling, bladders were pre-treated with the A2B receptor antagonist PSB 603 (500 nM). Preincubation with PSB 603 prevented the effect of severe hypoxia on transient pressure events (Fig. 5*B* and *C*), suggesting that hypoxic suppression of bladder activity requires A2B receptor activation.

The effects of severe hypoxia were examined using myh11-GCaMP6f mice, and $Ca^{2+}$ events prevalence and coincidence were quantified (Fig. 6*A*). Severe hypoxia potently reduced UBSM $Ca^{2+}$ prevalence and coincidence by 89.6% and 71.7%, respectively, an effect that was reversed by preincubation with the A2B receptor antagonist PSB 603 (Fig. 6*A* and *B*). Movie 4 shows representative effects of hypoxia, PSB 603 and PSB 603 + hypoxia on UBSM $Ca^{2+}$ activity.

## Discussion

The urinary bladder continuously fills, stores and voids urine. Under normal physiological conditions the spacing and frequency of these events are painless and well spaced to limit disruption to daily activities. However the mechanisms that regulate bladder contractility, sensation of fullness and bladder dysfunction are incompletely understood. This gap in knowledge limits our ability to pinpoint cellular or tissue-wide disruptions that occur during disease. Greater mechanistic insights into the physiological properties that govern bladder functions such as bladder relaxation/excitability are critical to understand how these processes change during acute and chronic disease conditions.

The principal aim of this study was to determine whether adenosine, a critical signalling molecule in other excitable tissues, plays a role in bladder contractility and to further explore downstream signalling targets that may be involved in this response. It further investigates whether acute hypoxia alters bladder activity and if adenosine signalling mediates potential hypoxia-driven responses. Here we demonstrate that adenosine exerts a potent relaxing effect on transient bladder contractility and that this response is mediated through A2B receptor stimulation and downstream activation of $BK_{Ca}$ channels. We also demonstrate that acute exposure to nearly anoxic conditions (0.1%–0.2% $O_2$, hypoxia) eliminates transient bladder activity and that this response relies on activation of A2B receptors.

Local release of adenosine and subsequent purinergic signalling pathways involving A1, A2 and A3 families of adenosine receptors play a critical role in regulating excitability in the nervous system and within striated and smooth muscle containing tissues such as heart, skeletal muscle and gastrointestinal tissues (Karmouty-Quintana et al., 2013; Mubagwa & Flameng, 2001; Sebastiao & Ribeiro, 2009; Zheng et al., 2007). Others have demonstrated that genetic ablation of A2B receptor function in mice resulted in an overactive bladder phenotype, highlighting the important negative feedback or relaxing roles these receptors may play in whole animal voiding behaviour (Hao et al., 2019).

In our experiments preincubation with an A2B receptor antagonist did not alter bladder contractility. This suggests that A2B receptors are not active under basal conditions. However this does not rule out the

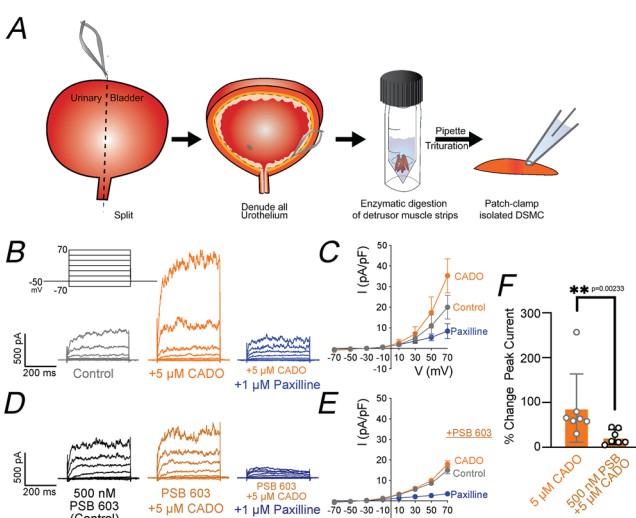

**Figure 4. Adenosine enhances whole-cell large-conductance calcium-activated potassium channel currents in UBSM (urinary bladder smooth muscle) cells**
*A*, illustration demonstrating the process for freshly isolated UBSM cells from whole bladder. *B*, voltage clamp protocol, holding potential of –50 mV and 300 ms voltage steps from –70 to +70 mV (20 mV steps). Representative traces using whole-cell configuration with control (normal bath), 2-chloroadenosine (CADO, 5 µM) and paxilline (1 µM) conditions. *C*, summarized current–voltage relationship under conditions described in (*B*). *D*, representative traces using whole-cell configuration with PSB 603 containing bath solution (500 nM), CADO (5 µM) and paxilline (1 µM) conditions. *E*, summarized current–voltage relationship under conditions described in (*D*). *F*, summary data of increase in peak current (at +70 mV) in UBSM during CADO application without and with (Hao et al., 2019) A2B receptor antagonist, PSB (PSB 603, 500 nM). Data shown are mean ± SD; *n* = 7 cells per group in (*F*) from *N* = 5 animals; individual *P*-values shown using Kruskal–Wallis test.

possibility of constitutive adenosine release and A2B receptor stimulation under basal conditions *in vivo*. Unlike the *in vivo* bladder the *ex vivo* bladder lacks vascular delivery of oxygen, glucose and other substances, as well as input from efferent nerve terminals. *In vivo* the bladder is subject to dynamic fluctuations in oxygen and glucose delivery and neurotransmitter input. Indeed Hao et al. observed dramatic alterations in mouse voiding behaviour when A2B receptor function was ablated (Hao et al., 2019).

The role of $K^+$ channels, key downstream targets of GsPCR-cAMP-PKA signalling (e.g. A2B receptor), remains unexplored in adenosine-induced bladder relaxation. Because $K_{ATP}$ channels in smooth muscle are primarily activated by PKA activity (Brayden, 2002; Kleppisch & Nelson, 1995; Quinn et al., 2004) with many reports of functional $K_{ATP}$ channel expression in UBSM (Bonev & Nelson, 1993; Malysz & Petkov, 2020; Petkov et al., 2001), these channels were a logical target to explore in the context of GsPCR-cAMP-PKA signalling via A2B receptors. Indeed synthetic activation of $K_{ATP}$ channels in this study and others (Petkov et al., 2001) provided a potent relaxing effect on the bladder that was inhibited with the $K_{ATP}$ channel blocker glibenclamide (Fig. 3*B*). Interestingly the potent relaxing effects of adenosine were unaltered in presence of $K_{ATP}$ channel blockade with glibenclamide (Fig. 3*D*). In other tissues such as blood vessels, the hyperpolarizing effect of adenosine signalling is tightly linked to $K_{ATP}$ channel activity (Kleppisch & Nelson, 1995; Sancho et al., 2022). These findings suggest either that the $K_{ATP}$ channel in UBSM is decoupled from A2B receptor-specific PKA activity or that the Kir6.X and SUR isoforms that comprise UBSM $K_{ATP}$ channels are distinct from other smooth muscle in their sensitivity (or lack thereof) to PKA. Others have reported that $K_{ATP}$ channels in porcine bladder smooth muscle are comprised of Kir 6.1 and SUR2A and further demonstrated that SUR2A is explicitly insensitive to phosphorylation by PKA (Kajioka et al., 2008). Both intriguing possibilities warrant further investigation, as physiological conditions or endogenous ligands that may activate UBSM $K_{ATP}$ channels are unknown.

Bladder smooth muscle $K_{ATP}$ channels exhibited no apparent sensitivity to adenosine. However PKA can also increase the activity of smooth muscle $BK_{Ca}$ channels. The UBSM $BK_{Ca}$ channel plays a critical role in regulating bladder excitability due to its outsized contribution to UBSM action potential repolarization. Particularly even slight increases in $BK_{Ca}$ channel activity within the UBSM physiological membrane potential range of –50 to +10 mV are sufficient to suppress smooth muscle contractility (Sancho & Kyle, 2021; Schubert & Nelson, 2001). Based on this we investigated whether A2B receptor stimulation enhances $BK_{Ca}$ channel activity in UBSM.

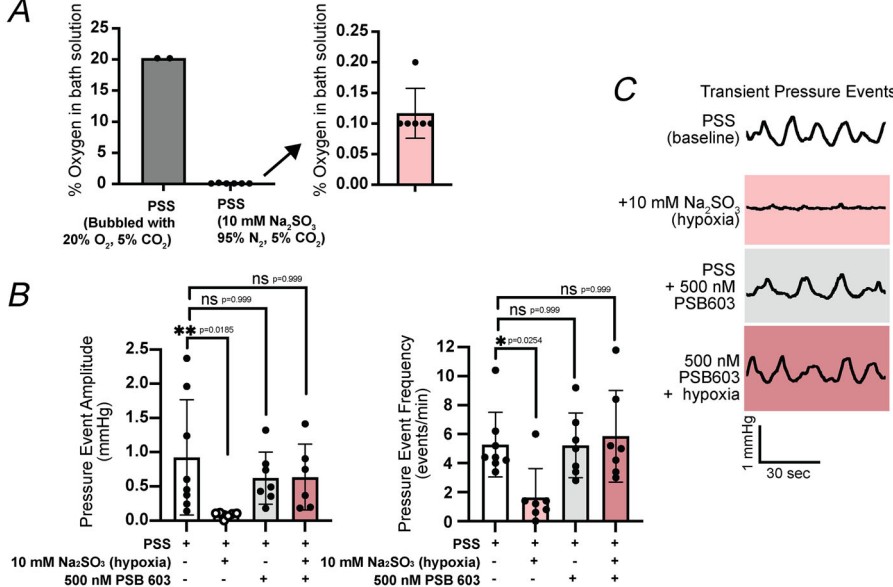

**Figure 5. Hypoxic conditions reduce whole bladder transient pressure event amplitude and frequency in an A2B receptor-dependent manner**

*A*, measured percentage concentration of oxygen in PSS and chemical hypoxic conditions (PSS with 10 mM $Na_2SO_3$ bubbled with 95% $N_2$ and 5% $CO_2$). *B*, summarized effects of physiological salt solution (PSS, baseline condition), hypoxic conditions ($Na_2SO_3$, 10 mM) and A2B receptor antagonist PSB 603 (500 nM). Data generated during a 5-min interval after at least 15 min of drug/compound exposure. *C*, representative pressure traces during conditions from (*B*). For (B) data shown are mean ± SD; *N* = 6–11 per group; individual *P*-values shown using Kruskal–Wallis test unless ***$P < 0.001$.

Because $K_{ATP}$ channels had no apparent sensitivity to adenosine and smooth muscle $BK_{Ca}$ channels are also strongly modulated by PKA activity, we investigated whether UBSM $BK_{Ca}$ activity was enhanced after A2B receptor stimulation. The first indication these channels may be activated by adenosine was highlighted by the apparent lack of adenosine-induced relaxation in bladders pretreated with a $BK_{Ca}$ channel inhibitor (Fig. 3*F*). We further investigated $BK_{Ca}$ channel sensitivity to adenosine receptor stimulation using conventional whole-cell patch-clamp electrophysiology and freshly isolated UBSM from mice. Indeed adenosine potently increased paxilline-sensitive outward currents, an effect that was sensitive to pretreatment with the A2B receptor antagonist PSB 603 (Fig. 4*B* and *D*). This study did not examine the exact mechanism leading to increased $BK_{Ca}$ channel activity downstream of A2B receptor stimulation. Potential mechanisms include PKA phosphorylation of $BK_{Ca}$ C-terminal residues, thus shifting the voltage and calcium sensitivity of the channel (Shipston & Tian, 2016;

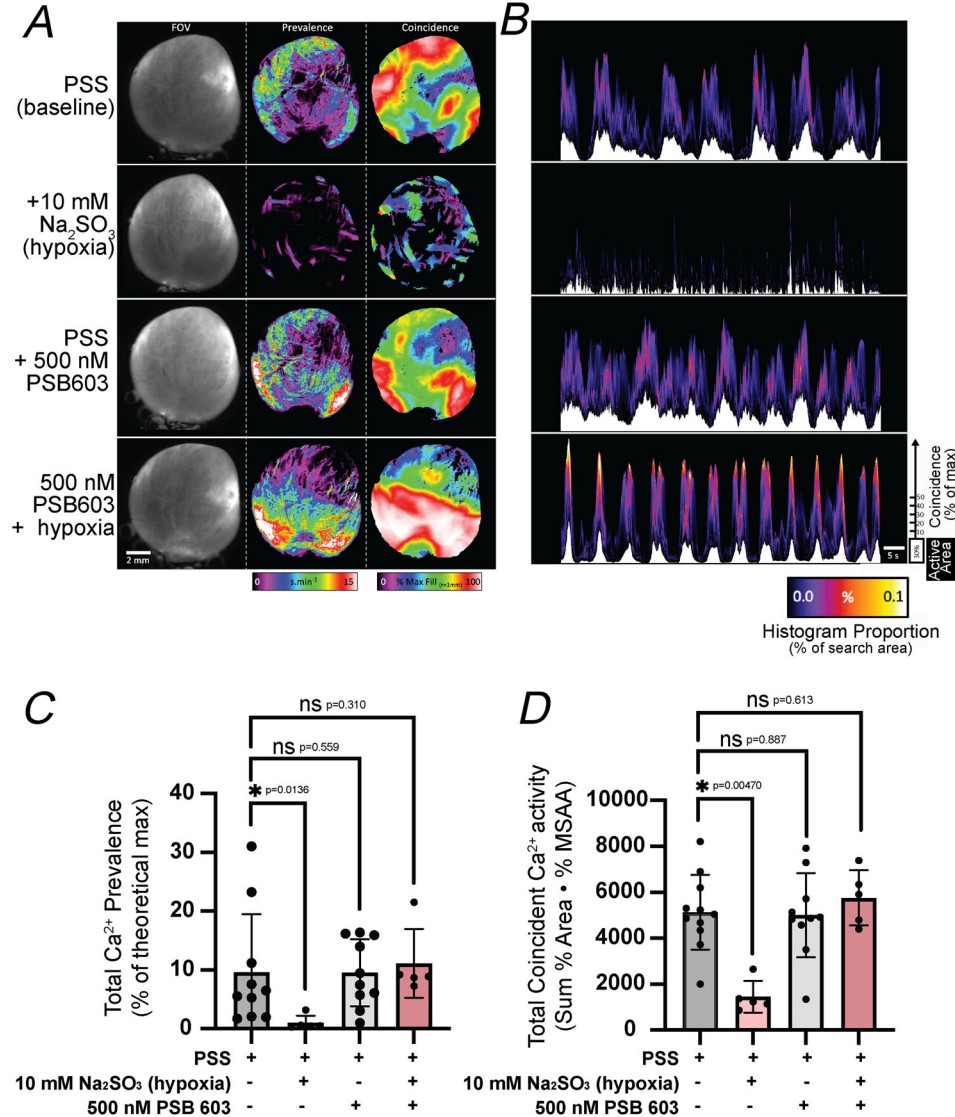

**Figure 6. Hypoxic conditions reduce urinary bladder smooth muscle calcium activity in an A2B receptor-dependent manner**
*A*, representative experiment showing FOV (field of view) (left) and parameter overlays of prevalence and maximum coincidence. *B*, spatio-temporal maps of combined prevalence (white trace: bottom) and coincidence (proportion of coincidence values mapped as cold/hot, with the actual coincidence value from 0% to 100% plotted on the *y*-axis superimposed on top of the prevalence trace). *C*, summarized effects of physiological salt solution (PSS, baseline condition), hypoxic conditions ($Na_2SO_3$, 10 mM) and A2B receptor antagonist PSB 603 (500 nM) on prevalence. *D*, summarized effects of conditions from (*B*) on coincidence. Data shown are mean ± SD; *N* = 5–10 per group; individual *P*-values shown using Kruskal–Wallis test unless ***$P < 0.001$.

Tian et al., 2004) or PKA-mediated increased ryanodine receptor spark activity (Jaggar et al., 2000; Porter et al., 1998; Wehrens et al., 2006).

Another important aspect of adenosine signalling, which is not addressed in this current study, is the endogenous source of adenosine within the bladder. In other tissues such as the brain, neurons are primarily exposed to adenosine from adjacent astrocytes (Haydon & Carmignoto, 2006). The urothelium, which serves as the inner lining of the bladder and abuts the detrusor through an interstitial interface, is a rich site for ATP (Andersson, 2015) and adenosine via hydrolysis. Indeed there are many observations and potential implications for ATP release from urothelium onto UBSM and bladder nerve fibres (Andersson, 2015; Merrill et al., 2016). Furthermore the interstitial side of the urothelium contains rich expression of functional ectonucleotides that rapidly convert ATP to adenosine within the space linking urothelium and detrusor layers (Durnin et al., 2019; Gutierrez Cruz et al., 2022). Although the urothelium is a logical source of adenosine, further studies are required to definitively link adenosine release to urothelial cells during physiological or pathological stimuli.

Finally we used severe hypoxic conditions to investigate whether urinary bladder adenosine signalling is engaged in the context of hypoxia. Hypoxic or ischaemic conditions in the urinary bladder are a putative under-lying phenomenon in many pathologies of the lower urinary tract, including overactive and underactive bladder disorders, benign prostate hypertrophy, bladder cancer and urge disorders associated with bladder nerve dysfunction (Andersson et al., 2017; Lodhi et al., 2021; Shimizu et al., 2014; Zhao et al., 2016). Furthermore in other excitable tissues such as the brain and heart, there is an established link between low oxygen conditions and adenosine signalling, where locally released adenosine in response to hypoxia and subsequent adenosine receptor stimulation plays a critical protective role (Haydon & Carmignoto, 2006; Mubagwa & Flameng, 2001). Although our model reflects acute and severe hypoxia, we reasoned that it offers a controlled platform to investigate fundamental signalling mechanisms in a reproducible and robust manner. The 15–25 min acute hypoxia treatment does not appear to directly impair ATP-sensitive processes. However prolonged exposure to low oxygen conditions would be expected to eventually reduce cytosolic ATP levels, leading to dysfunction in ATP-dependent mechanisms such as contractile protein function. Our findings indicate that A2B receptor stimulation is critical for the relaxing effects of severe chemical hypoxia, supporting a direct link between hypoxia and adenosine signalling in the urinary bladder.

In whole animal models and in clinical studies, hypoxia or ischaemia is associated with both overactive and underactive bladder phenotypes (Andersson et al., 2017; Azadzoi et al., 1999; Zhao et al., 2016). This seemingly disparate effect on bladder contractility may reflect the timing and duration of hypoxic condition in the urinary bladder and potential compensation by distinct pathways within various bladder cell types. Furthermore other ischaemic factors, such as peptide signalling, low glucose and reactive oxygen species, may shape bladder pathology beyond the effects of hypoxia alone. Nevertheless our observation of complete reversal of hypoxia-driven bladder relaxation through A2B receptor inhibition positions this pathway as an attractive target for treating bladder pathologies associated with hypoxic conditions.

## Methods

### Animals

**Pressurized bladder preparations.** The urinary bladder, ureters and urethra were excised from a mouse and placed in an ice-cold HEPES dissecting solution consisting of 134 mM NaCl, 6 mM KCl, 1 mM $MgCl_2$, 2 mM $CaCl_2$, 10 mM HEPES and 7 mM glucose (pH 7.4). The ureters were tied close to the bladder wall using 10.0 suture. The *ex vivo* bladder preparation was then placed in a specialized recording chamber and superfused with bicarbonate-buffered PSS consisting of 118.5 mM NaCl, 4.6 mM KCl, 1.2 mM $KH_2PO_4$, 1.2 mM $MgCl_2$, 2 mM $CaCl_2$, 24 mM $NaHCO_3$ and 7 mM glucose; the pH of the solution was maintained at 7.4 by bubbling with biological atmosphere gas (20% $O_2$, 5% $CO_2$, balance $N_2$). All experiments were performed at 37°C. A cannula with three arms was used to infuse saline into the bladder and empty the bladder. One arm of the cannula was inserted through the urethra into the bladder lumen and ligated in place using 10.0 suture. The second arm of the cannula was attached to a pressure transducer and syringe pump to measure intravesical pressure and infuse saline into the bladder lumen, respectively. The third arm of the cannula was used to manually empty the bladder when bladder pressure reached 26 mmHg. Saline was infused using a syringe pump at a rate of 30 µl/min. Bladder pressure was measured using a pressure transducer (PT-F, Living Systems Instrumentation, Saint Albans, VT, USA) connected to a signal conditioner (model NL-108, Digitimer, Hertfordshire, UK) set to 100 mV/cmH₂O output for recording bladder pressure. A Power3A analog–digital converter and Spike2 software (Cambridge Electronic Design, Cambridge, UK) was used to record data at a rate of 100 samples per second. Transient pressure events were analysed offline with Spike2 software using peak detection features with an amplitude threshold of 0.05 mmHg (apply smoothing to pressure signal with a time constant of 0.5 s followed by Spike2 'Peak Find' function).

Chemical hypoxia was achieved by adding 10 mM $Na_2SO_3$ to bicarbonate-buffered PSS and bubbled with 95% $N_2$ and 5% $CO_2$. Hypoxic conditions were added to isolated bladder in 15–25 min bouts. Oxygen levels were measured using a daily calibrated inline oxygen probe (Flow-thru Oxygen Electrode, Microelectrodes, Inc., Bedford, NH, USA).

## Widefield $Ca^{2+}$ imaging and analysis

**Imaging equipment.** $Ca^{2+}$-induced fluorescence in detrusor smooth muscle of the *myh11*-GCaMP6f urinary bladder was visualized on an Olympus MVX10 Macroview microscope (1.0× PLANAPO objective) using X-Cite Xylis light source with Chromus EGFP excitation/emission filter set (ET470/40x, T495lpxr, ET525/50m, catalogue no. 49002). Fluorescence was captured using an Andor Zyla 4.2 CMOS camera ($2048 \times 2048 \times 16$bit (binned to $1024 \times 1024$)) and micro-manager 1.4 software (Edelstein et al., 2010, 2014) for 80 s at 25.1 frames per second (fps).

**Movie preprocessing.** Movies (4.2 Gb) were imported into ImageJ, and the bladder was cropped from extraneous non-bladder background (2–3 Gb). A debleaching routine was then used to counter dimming of fluorescence during recordings (linear or exponential) followed by a deflickering routine that corrected sharp jumps in brightness due to unstable light sources or room conditions. Due to the single attachment point at the cannula positioned in the urethra, the dome of the bladder could tilt during contractions. By tracking a point on the dome and referencing the angle to the cannula, the movie was rotated to a set angle to correct tilting motions (angular normalization). Motions that resulted in slight displacement were also corrected (XY dolly normalization).

**$Ca^{2\pm}$ extraction.** Movies were then filtered (Gaussian blur: $3 \times 3$ pixels, SD = 1.0) to reduce granular shot noise. No temporal filtering was used. A modified standard deviation of quiescence (SDqe) routine (Heppner et al., 2019; Longden et al., 2021) was used to demarcate pixels in which fluorescence was elevated above background fluctuations in intensity. The values used were as follows: $SD_{min} = 6.0$ and $SD_{threshold} = 2.1$ with a quiescence estimator (QE) between 15% and 25%. Extracted particles representing $Ca^{2+}$ events were size filtered ($>35$ pixel area: $\sim0.12 \times 0.12$ mm in size) and saved as co-ordinate-based spatio-temporal objects.

**$Ca^{2\pm}$ event refinement.** Non-uniform $Ca^{2+}$ events caused contractions and bulges and indentions of the wall of the bladder that was particularly observed around the outer perimeter. Similarly $Ca^{2+}$ events at the outer perimeter were much brighter due to the greater volume of the bladder wall at the edges when projected onto a flat plane. The outer 5–10° (corresponding to $\sim0.2$–0.5 mm depending on the size of the bladder) was masked out, thereby ensuring extracted $Ca^{2+}$ events were from areas largely free from edge volume and wall motion artefacts. Regardless $Ca^{2+}$ events occurring in large regions of detrusor evoked contractions that distorted the bladder wall causing spatial movement artefacts to occur. Due to the small size of these distortion artefacts, they could be filtered out without affecting $Ca^{2+}$ events in detrusor smooth muscle cells (particle size $>100$–150 pixels, $\sim0.2 \times 0.2$ μm. See movie 1, middle panel, blue particles). Finally floating debris in the field of view (FOV) that coursed over the bladder could be removed by deleting trajectories of bright particles that were observed beyond the outer edge of the bladder wall during the recording.

## $Ca^{2\pm}$ event measurements: amplitude

We chose not to use measurements of the amplitude of detrusor $Ca^{2+}$ activity due to the wall thickness issues referred to earlier around the outer perimeter of the bladder. The predominant type of $Ca^{2+}$ events observed in these experiments corresponds to 'muscle action potentials' that have relatively uniform amplitudes at low to moderate frequencies of firing. Measuring the frequency, duration and area of these $Ca^{2+}$ events provides a more structured and unbiased appreciation of their effect on bladder motility than amplitude *per se*.

## $Ca^{2\pm}$ event measurements: prevalence

To best summarize the overall amount of $Ca^{2+}$ activity throughout the entire visible hemisphere of isolated bladders during the 80-s recordings, a measure of $Ca^{2+}$ event prevalence was calculated. This measure accumulates the area and duration of extracted $Ca^{2+}$ events which can be mapped onto the bladder surface (Figs 2 and 6) or condensed into a single value by expressing prevalence in relation to the maximal theoretical $Ca^{2+}$ activity (100% bladder area multiplied by movie duration; see Movie 1, rightmost panel).

## $Ca^{2\pm}$ event measurements: coincidence

Prevalence measurements do not offer any insight into the patterning of $Ca^{2+}$ activity – only the overall amount generated at every point on the bladder surface. Questions such as (1) does $Ca^{2+}$ activity occur synchronously in regions of the bladder or is it confined to individual cells? and (2) does $Ca^{2+}$ activity propagate? can be answered only if there is a measurement that monitors $Ca^{2+}$ activity

in adjacent UBSM fibres. We adapted Euclidian distance mapping (EDM) routines to radially scan within and around each pixel of each active $Ca^{2+}$ event to gauge the proportion of nearby cells that were coincidently active. For this study we used a planar (XY) search radius of 0.96 mm and no additional temporal tolerance, such that coincident activity had to occur in adjacent UBSM fibres in exactly the same frame as the reference $Ca^{2+}$ activity. As most $Ca^{2+}$ events had durations >1 frame (40 ms), slight delays in the activation of neighbouring bundles were accommodated. Raw coincidence results were normalized to maximum area of active pixels experienced around each reference pixel in the search window during the entire recording and are expressed as the percentage of maximum search area activity within a radius of 1 mm ($\%MSAA_{r = 1\,mm}$). Although coincidence values were calculated for every pixel of every $Ca^{2+}$ event in every frame of each recording, we chose to present only the highest coincidence value that occurred at every point on the bladder surface ($Max\%MSAA_{r = 1\,mm}$: Figs 4 and 6). Temporal characteristics of coincidence were plotted as spatio-temporal maps showing overall prevalence (white areas at the bottom of each map), with the proportion of coincidence values from the bladder surface from each frame converted to a histogram and superimposed on top of the prevalence values. The fire colouring represents the percentage of each coincidence value from the bladder surface, and the vertical position in the *y*-axis represents the %MSAA value ranging from 0 (at the intersection of the underlying white prevalence plot) to 100 (all cells in the search radius were active at the same time: towards the top of each Spatio-Temporal (ST) map). For example the third (PSS + PSB 603) and fourth (PSB + hypoxia) panels in Fig. 6*A* show similar prevalence values throughout the bladder, but the pattern of $Ca^{2+}$ activity after hypoxia shows much higher levels of coincidence, particularly in the trigone half of the hemisphere (red/white band in the bottom half of the image). The temporal dynamics of $Ca^{2+}$ activity in the ST maps after hypoxia reveal regular, ongoing bouts of $Ca^{2+}$ activity associated with a high proportion (yellow/white colour) of highly coincident (towards the top of the *y*-axis of the map) $Ca^{2+}$ activity, consistent with the coherent spread of $Ca^{2+}$ waves.

### Smooth muscle isolation and patch-clamp electrophysiology

The urinary bladder was removed and placed in an ice-cold HEPES cell dissociation solution consisting of 55 mM NaCl, 5.6 mM KCl, 2 mM $MgCl_2$, 80 mM Na-glutamate, 10 mM glucose and 10 mM HEPES (pH 7.3). The detrusor layer was removed and cut into six to eight strips. The strips were incubated in papain (2 mg/ml) and 1,4-dithioerythritol (DTE, 2 mg/ml) dissolved in the dissociation solution for 20 min at 37°C. Strips were thoroughly washed with ice-cold dissociation solution and placed in collagenase type H (2 mg/ml, dissolved in dissociation solution) for 6 min at 37°C. The strips were thoroughly washed with ice-cold dissociation solution and then triturated with a fire-polished Pasteur pipette to release single dissociated smooth muscle cells (2 ml final volume in dissociation solution). Cells were stored on ice for up to 6 h. An aliquot of cells (500 μL) was added to 1 ml of cell dissociation solution in a custom electrophysiology chamber with a glass coverslip bottom at room temperature ($\sim$22°C). Aliquoted cells were undisturbed for 30–50 min. Once adhered cells were washed with bath solution consisting of 134 mM NaCl, 6 mM KCl, 1 mM $MgCl_2$, 2 mM $CaCl_2$, 7 mM glucose and 10 mM HEPES (pH 7.4). Patch pipettes were pulled using a borosilicate glass (1.5 mm outer diameter and 1.1 mm inner diameter) with filament (Sutter Instruments, Novato, CA, USA) and then fire polished to a tip resistance of 3.5–5 MΩ. Patch pipettes were filled with pipette (intracellular) solution consisting of 107 mM KCl, 33 mM KOH, 5 mM NaCl, 1.1 mM $MgCl_2$, 5 mM EGTA, 3.2 mM $CaCl_2$ ($\sim$300 nM free $Ca^{2+}$) and 10 mM HEPES (pH 7.2). The pipette was gently manoeuvred onto the UBSM cell using a mechanical micromanipulator (MP-285, Sutter Instruments). A high-resistance gigaohm (>1 GΩ) seal was made on the UBSM cell membrane using slight negative pressure, and then the membrane was ruptured using rapid negative pressure. Whole-cell capacitance and series access resistance were measured using the cancellation circuitry on the amplifier. The voltage step protocol had a holding potential of –50 mV, with 20 mV depolarizing steps (300 ms) from –70 to +70 mV. Whole-cell currents were recorded using an Axopatch 200B current amplifier (1 kHz filtering) and digitized at 5 kHz using 1322A Digidata and pClamp 9 software (amplifier, digitizer and pClamp software, Axon Instruments, Molecular Devices, San Jose, CA, USA). A total of 14 UBSM cells were recorded with a whole-cell capacitance of 44.04 ± 3.65 pF (mean ± SD).

### Statistics and analysis

Normality was not assumed because the sample size was <30 in all data sets (Curran-Everett, 2017). All comparisons were unpaired multiple comparisons performed using Kruskal–Wallis test or unpaired one-sample Wilcoxon test.

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

## Additional information

### Data availability statement

All individual data used to generate figures, perform statistical analysis and inform overall conclusions are found in respective summary data graphs. Detailed methodology to attain SD-based $Ca^{2+}$ analysis and transient pressure analysis can be found in previously published articles (Heppner et al., 2016; Longden et al., 2021).

### Competing interests

G.M.H. is a scientific consultant for MED Associates, Inc., and Living Systems Instrumentation, a division of Catamount Research and Development, Inc., and his wife is a co-owner of these companies. All other authors have no disclosures.

## Author contributions

N.R.K. and G.M.H. designed the experiments and conceptualized the project. N.R.K., G.M.H., T.J.H., J.L.R., M.S., A.M.H. and S.H.C. conducted the experiments. G.W.H. developed $Ca^{2+}$ analysis methodologies. N.R.K., G.M.H., T.J.H., G.W.H., J.L.R. and M.S. analysed data. N.R.K., G.M.H. and M.T.N. provided resources and animal models. N.R.K., G.W.H. and G.M.H. wrote the manuscript.

## Funding

This work was supported by the National Institute of Diabetes and Digestive and Kidney Diseases Grant R01DK125543 (to G.M.H. and T.J.H.) and Grant F99DK143563 (to J.L.R.), the National Institute of General Medical Sciences P20-GM-135007 (to M.T.N., Customized Physiology and Imaging Core Support to G.M.H., T.J.H. and G.W.H., and Project Director and Pilot Project Support to N.R.K.) and the American Heart Association Postdoctoral Fellowship 24POST1188081 (to S.H.C.).

## Acknowledgements

The authors thank H. Ryan, T. Wellman, D. Enders and N. Cashen for technical assistance and animal care.

## Keywords

adenosine receptor, calcium-activated potassium channel, hypoxia, urinary bladder

## Supporting information

Additional supporting information can be found online in the Supporting Information section at the end of the HTML view of the article. Supporting information files available:

**Peer Review History**
**Movie 1**
**Movie 2**
**Movie 3**
**Movie 4**

