## [Peer Review History · The Journal of Physiology]

Adenosine and acute low oxygen conditions suppress urinary bladder contractility through the activation of A2B receptors and BKCa channels

Gerald M. Herrera, Jason L Rengo, Grant W. Hennig, Thomas Heppner, Alexandria M Hepp, Maria Sancho, Saul Huerta de la Cruz, Mark T Nelson, and Nicholas R. Klug

DOI: 10.1113/JP289080

Corresponding author(s): Nicholas Klug (nicholas.klug@uvm.edu)

The following individual(s) involved in review of this submission have agreed to reveal their identity: Caoimhin Sean Griffin (Referee #1); Hikaru Hashitani (Referee #2)

Review Timeline:

Submission Date:	16-Apr-2025
Editorial Decision:	08-May-2025
Revision Received:	16-Sep-2025
Editorial Decision:	25-Sep-2025
Revision Received:	25-Sep-2025
Accepted:	28-Sep-2025

Senior Editor: Peking Fong

Reviewing Editor: Bernard Drumm

Transaction Report:

Dear Dr Klug,

Re: JP-RP-2025-289080 "Adenosine and acute low oxygen conditions suppress urinary bladder contractility through the activation of A2B receptors and BKCa channels" by Gerald M. Herrera, Thomas Heppner, Grant W. Hennig, Jason L Rengo, Alexandria M Hepp, Maria Sancho, Saul Huerta de la Cruz, Mark T Nelson, and Nicholas R. Klug

Thank you for submitting your manuscript to The Journal of Physiology. It has been assessed by a Reviewing Editor and by 2 expert referees and we are pleased to tell you that it is potentially acceptable for publication following satisfactory major revision.

LANGUAGE EDITING AND SUPPORT FOR PUBLICATION: If you would like help with English language editing, or other article preparation support, Wiley Editing Services offers expert help, including English Language Editing, as well as translation, manuscript formatting, and figure formatting at www.wileyauthors.com/eoo/preparation. You can also find resources for Preparing Your Article for general guidance about writing and preparing your manuscript at www.wileyauthors.com/eoo/prepresources.

REVISION CHECKLIST:

We look forward to receiving your revised submission.

Yours sincerely,

Peying Fong
Senior Editor
The Journal of Physiology

REQUIRED ITEMS

- Include a Key Points list in the article itself, before the Abstract.

- Author photo and profile. First or joint first authors are asked to provide a short biography (no more than 100 words for one author or 150 words in total for joint first authors) and a portrait photograph. These should be uploaded and clearly labelled together in a Word document with the revised version of the manuscript. See Information for Authors for further details.

- You must start the Methods section with a paragraph headed Ethical approval (https://jp.msubmit.net/cgi-bin/main.plex?form_type=display_requirements#methods).

Research must comply with The Journal's policies regarding animal experiments (<https://physoc.onlinelibrary.wiley.com/hub/animal-experiments>) and adherence to these policies must be stated in the manuscript.

Authors should confirm in their Methods section that their experiments were carried out according to the guidelines laid down by their institution's animal welfare committee, including an ethics approval reference number. The Methods section must contain a statement about access to food, water and housing, details of the anaesthetic regime: anaesthetic used, dose and route of administration, and method of killing the experimental animals.

- The reference list must be in alphabetical order, rather than numbered, to comply with our Journal format.

- Please ensure that the Article File you upload is a Word file.

- Your paper contains Supporting Information of a type that we no longer publish, including supplementary tables and figures. Any information essential to an understanding of the paper must be included as part of the main manuscript and figures. The only Supporting Information that we publish are video and audio, 3D structures, program codes and large data files. Your revised paper will be returned to you if it does not adhere to our Supporting Information Guidelines.

- Papers must comply with the Statistics Policy: https://jp.msubmit.net/cgi-bin/main.plex?form_type=display_requirements#statistics.

In summary:

- If n {less than or equal to} 30, all data points must be plotted in the figure in a way that reveals their range and distribution. A bar graph with data points overlaid, a box and whisker plot or a violin plot (preferably with data points included) are acceptable formats.
- If $n > 30$, then the entire raw dataset must be made available either as supporting information, or hosted on a not-for-profit repository, e.g. FigShare, with access details provided in the manuscript.
- 'n' clearly defined (e.g. x cells from y slices in z animals) in the Methods. Authors should be mindful of pseudoreplication.
- All relevant 'n' values must be clearly stated in the main text, figures and tables.
- The most appropriate summary statistic (e.g. mean or median and standard deviation) must be used. Standard Error of the Mean (SEM) alone is not permitted.
- Exact p values must be stated. Authors must not use 'greater than' or 'less than'. Exact p values must be stated to three significant figures even when 'no statistical significance' is claimed.

- A Data Availability Statement is required for all papers reporting original data. This must be in the Additional Information section of the manuscript itself. It must have the paragraph heading 'Data Availability Statement'. All data supporting the results in the paper must be either: in the paper itself; uploaded as Supporting Information for Online Publication; or archived in an appropriate public repository. The statement needs to describe the availability or the absence of shared data. Authors must include in their statement: a link to the repository they have used, or a statement that it is available as Supporting Information; reference the data in the appropriate section(s) of their manuscript; and cite the data they have shared in the References section. Whenever possible, the scripts and other artefacts used to generate the analyses presented in the paper should also be publicly archived. If sharing data compromises ethical standards or legal requirements then authors are not expected to share it, but must note this in their statement. For more information, see our Statistics Policy.

- Please include an Abstract Figure file, as well as the Figure Legend text within the main article file. The Abstract Figure is a piece of artwork designed to give readers an immediate understanding of the research and should summarise the main conclusions. If possible, the image should be easily 'readable' from left to right or top to bottom. It should show the physiological relevance of the manuscript so readers can assess the importance and content of its findings. Abstract Figures should not merely recapitulate other figures in the manuscript. Please try to keep the diagram as simple as possible and without superfluous information that may distract from the main conclusion(s). Abstract Figures must be provided by authors no later than the revised manuscript stage and should be uploaded as a separate file during online submission labelled as File Type 'Abstract Figure'. Please also ensure that you include the figure legend in the main article file. All Abstract Figures should be created using BioRender. Authors should use The Journal's premium BioRender account to export high-resolution images. Details on how to use and access the premium account are included as part of this email.

EDITOR COMMENTS

Reviewing Editor:

Thank you for your submission.

Both reviewers have highlighted the potential insights derived from the whole bladder imaging techniques described in the manuscript and its usefulness in elucidating the mechanisms of purinergic signalling in the bladder. However both reviewers have commented that the study needs expanding with additional data in order to maximise the impact of this paper. In the case of reviewer 1, they point to the need for sufficient biological replicates in the experiments already presented in order to reach definitive conclusions on the provided data. Reviewer 2 highlights the need for an extension of the description of the whole bladder calcium events analysis and how this is differentiated from contractile events.

Please also see 'Required Items' above.

Senior Editor:

Comments for Authors to ensure the paper complies with the Statistics Policy (Required):

Please note that the published Statistics Policy states that errors must be presented as standard deviation rather than

standard error of the mean. Moreover, exact p values must be provided to three significant figures (not significant digits).

Comments to the Author:

Initial review of your manuscript, "Adenosine and acute low oxygen conditions suppress urinary bladder contractility through the activation of A2B receptors and BKCa channels" is now complete. As you will read from the attached comments, both Expert Referees find the topic and approach to carry potential impact. However, as summarized by the Reviewing Editor, both also suggest significant additional experimentation is needed before this study can achieve this potential fully. The comments of both are detailed to a sufficient extent that you can prioritize the additional work that must be performed. Please pay particular attention to Referee 2's comments regarding the implications of your data interpretation.

In addition to the comments raised by Referee 1 pertaining to statistical powering, please review The Journal of Physiology's Statistical Policy when preparing your data for resubmission. Specifically, data variation needs to be expressed as standard deviation rather than standard error of the mean. Also, when presenting tests of significance, exact p values to three significant figures (not decimal places) are required in cases (except when $p < 0.001$).

With regard to policy on Supporting Information files, note also that data currently presented as Supplemental figures 1 and 2 need to be incorporated within figures appearing within the manuscript proper.

Please also provide the animal protocol number under the "Animals" section within the Methods.

REFEREE COMMENTS

Referee #1:

This manuscript presents original work from Herrera et al., that investigates how adenosine signaling shapes normal and hypoxic urinary-bladder function. By combining an ex-vivo pressurised mouse-bladder preparation with single-cell patch-clamp recordings from detrusor smooth-muscle cells, the authors demonstrate a link between A2B adenosine-receptor activation and BK channels, which serves to dampen non-voiding pressure transients, detrusor Ca^{2+} flashes and overall bladder excitability.

The study is timely, because the field still lacks a clear mechanism linking purinergic tone, oxygen tension and detrusor contractility—a gap that is clinically relevant to overactive- and underactive-bladder syndromes as well as ischaemic bladder dysfunction.

Overall, the data add substantial new insight into the molecular mechanisms that regulate bladder excitability and offer new opportunities for novel therapeutic strategies that target A2B receptors as treatments for bladder pathologies.

Please, find below some points that should be addressed:

Major revision.

Figures 1, 2, 5, and 6 each include groups with only $N = 3$ experimental replicates, while Figure 4 reports $N = 4$. These sample sizes provide limited statistical power and insufficient biological breadth to support definitive conclusions. I therefore recommend increasing the number of independent experiments to at least N {greater than or equal to} 5 per group throughout the manuscript. If additional data has already been collected, please incorporate them; otherwise, further experiments will be required to meet this standard. Please revise the figure panels to display additional experiments/data points and update each panel's legend to reflect the updated sample sizes (revised N values).

Minor revisions.

- Please revise the schematic in Figure 3A so that Paxilline is depicted as a BK channel inhibitor/blocker, not an activator as it is currently illustrated.

- In Panel 3E, the displayed trace suggests that Paxilline slightly lowers event frequency from control conditions, yet the summary data indicates an increase. Please replace this trace with one that accurately represents the averaged results.

- In Figure 4B, the electrophysiological traces seem to show CADO alone in the second panel and Paxilline alone in the third. If the third panel actually represents the CADO + Paxilline condition, please relabel it clearly (e.g., "CADO + Paxilline") so readers can distinguish the treatments unambiguously.

- Lines 129-130: Please state explicitly that the phasic bladder contractions described here are cholinergic (Muscarinic) in origin.

Referee #2:

This study demonstrated that the adenosine analog CADO suppresses whole bladder transient pressure rises (TPRs) and whole bladder UBSM Ca²⁺ events through the activation of A2B adenosine receptor-mediated pathway where BK channels could be involved as a downstream effector. It was also shown that acute severe hypoxic conditions attenuate both TPRs and UBSM Ca²⁺ events via A2B adenosine receptor activation, highlighting the prominent role of A2B adenosine receptor in hypoxia-associated bladder contractile disorders. I found this study important to further our understanding of mechanisms underlying ischaemic bladder, but have several concerns about interpretation of the results as well as experimental design.

Major comments

1. Whole bladder UBSM Ca²⁺ imaging is a novel and well-designed technique that can be applied to understanding Ca²⁺ dynamics in the bladder wall. However, this technique does not seem to be effectively utilised in this study where UBSM Ca²⁺ signals are only shown as corresponding events of TPRs. Some basic information about the correlation or dissociation between the Ca²⁺ events and TPRs would be useful to demonstrate the relevance of this technique. Isn't it possible to measure the conducting velocity of the Ca²⁺ events or their synchrony amongst multiple sites? How can GCaMP signals be distinguished from contractile events, i.e., tissue distortions? Can the Ca²⁺ events be maintained in arrested bladders with wortmannin?. Showing the effects of nifedipine and paxilline alone would be helpful for the readers to realise that the Ca²⁺ events arise from LVDCC-dependent spontaneous action potentials.

2. Ln 180 (Fig.4). In mouse UBSM where iberiotoxin increases the amplitude of spontaneous action potentials and prolongs their repolarising phase, the peak value of normal action potentials is below +10mV (Hayase et al., (2009). J Urol. 181:2355-65). Does CADO significantly increase paxilline-sensitive outward currents at the physiological membrane potential range, e.g., between -50 mV and +10 mV? Ln 165 (Fig.3). Since paxilline alone should enlarge TPRs, adenosine may be capable of suppressing enlarged TPRs in paxilline-treated bladders. Therefore, effects of CADO on TPRs in bladders that had been pre-treated with paxilline should be tested.

Minor comments

1. Abstract Ln 49. Since the pharmacological blockade of A2B receptor with PSB 603 did not enhance TPRs (Fig.1B, C) or UBSM Ca²⁺ events (Fig.2B, C), it appears that adenosine is not continuously produced in the normal bladder to regulate physiological bladder contractility.

2. Ln 126 (Fig.S1). Since the TPRs arise from spontaneous UBSM contractions that are not sensitive to atropine, effects of A2A or A2B receptor agonists on spontaneous phasic contractions rather than CCh-induced oscillatory contractions should be examined.

3. Ln 196 (Fig.S2). Hypoxic conditions seem to be extreme. In guinea-pig bladder outlet obstruction model, the lowest oxygen saturation in the bladder wall even during voiding is about 40% (Scheepe et al. (2011), J Urol. 186:1128-33).

4. Ln 196. The complete restoration of TPRs (Fig.5A, B) or Ca²⁺ events (Fig.6A, B) upon the blockade of A2B adenosine receptor in hypoxic conditions is quite surprising. Even without the inhibitory actions of adenosine, UBSM contractions are expected to be diminished due to, for example, the severely reduced ATP production. How long were the bladders situated

in the hypoxic conditions?

5. Ln 196. Effects of paxilline on the hypoxia-induced suppression of TPRs or Ca²⁺ events should be tested. Alternatively, the last sentence in Abstract and corresponding descriptions in the main text should be rewritten.

END OF COMMENTS

This study demonstrated that the adenosine analog CADO suppresses whole bladder transient pressure rises (TPRs) and whole bladder UBSM Ca^{2+} events through the activation of A2B adenosine receptor-mediated pathway where BK channels could be involved as a downstream effector. It was also shown that acute severe hypoxic conditions attenuate both TPRs and UBSM Ca^{2+} events via A2B adenosine receptor activation, highlighting the prominent role of A2B adenosine receptor in hypoxia-associated bladder contractile disorders. I found this study important to further our understanding of mechanisms underlying ischaemic bladder, but have several concerns about interpretation of the results as well as experimental design.

Major comments

1. Whole bladder UBSM Ca^{2+} imaging is a novel and well-designed technique that can be applied to understanding Ca^{2+} dynamics in the bladder wall. However, this technique does not seem to be effectively utilised in this study where UBSM Ca^{2+} signals are only shown as corresponding events of TPRs. Some basic information about the correlation or dissociation between the Ca^{2+} events and TPRs would be useful to demonstrate the relevance of this technique. Isn't it possible to measure the conducting velocity of the Ca^{2+} events or their synchrony amongst multiple sites? How can GCaMP signals be distinguished from contractile events, i.e., tissue distortions? Can the Ca^{2+} events be maintained in arrested bladders with wortmannin?. Showing the effects of nifedipine and paxilline alone would be helpful for the readers to realise that the Ca^{2+} events arise from LVDCC-dependent spontaneous action potentials.
2. Ln 180 (Fig.4). In mouse UBSM where iberiotoxin increases the amplitude of spontaneous action potentials and prolongs their repolarising phase, the peak value of normal action potentials is below +10mV (Hayase et al., (2009). J Urol. 181:2355-65). Does CADO significantly increase paxilline-sensitive outward currents at the physiological membrane potential range, e.g., between -50 mV and +10 mV? Ln 165 (Fig.3). Since paxilline alone should enlarge TPRs, adenosine may be capable of suppressing enlarged TPRs in paxilline-treated bladders. Therefore, effects of CADO on TPRs in bladders that had been pre-treated with paxilline should be tested.

Minor comments

1. Abstract Ln 49. Since the pharmacological blockade of A2B receptor with PSB 603 did not enhance TPRs (Fig.1B, C) or UBSM Ca^{2+} events (Fig.2B, C), it appears that adenosine is not continuously produced in the normal bladder to regulate physiological bladder contractility.
2. Ln 126 (Fig.S1). Since the TPRs arise from spontaneous UBSM contractions that are not sensitive to atropine, effects of A2A or A2B receptor agonists on spontaneous phasic contractions rather than CCh-induced oscillatory contractions should be examined.
3. Ln 196 (Fig.S2). Hypoxic conditions seem to be extreme. In guinea-pig bladder outlet obstruction

model, the lowest oxygen saturation in the bladder wall even during voiding is about 40% (Scheepe et al. (2011), J Urol. 186:1128-33).

4. Ln 196. The complete restoration of TPRs (Fig.5A, B) or Ca^{2+} events (Fig.6A, B) upon the blockade of A2B adenosine receptor in hypoxic conditions is quite surprising. Even without the inhibitory actions of adenosine, UBSM contractions are expected to be diminished due to, for example, the severely reduced ATP production. How long were the bladders situated in the hypoxic conditions?
5. Ln 196. Effects of paxilline on the hypoxia-induced suppression of TPRs or Ca^{2+} events should be tested. Alternatively, the last sentence in Abstract and corresponding descriptions in the main text should be rewritten.

Herrera et al. Responses to Editors and Referees

Original text from Editors and Referees is in **bold dark blue**. Author's responses are found in black text.

Reviewing Editor:

Thank you for your submission.

Both reviewers have highlighted the potential insights derived from the whole bladder imaging techniques described in the manuscript and its usefulness in elucidating the mechanisms of purinergic signalling in the bladder. However both reviewers have commented that the study needs expanding with additional data in order to maximise the impact of this paper. In the case of reviewer 1, they point to the need for sufficient biological replicates in the experiments already presented in order to reach definitive conclusions on the provided data. Reviewer 2 highlights the need for an extension of the description of the whole bladder calcium events analysis and how this is differentiated from contractile events.

Please also see 'Required Items' above.

We sincerely thank the reviewing editor for their time and effort in handling our manuscript. We have addressed all 'Required Items' mentioned above with detailed attention on all journal specific guidelines regarding methods descriptions and statistical analysis and presentation.

Senior Editor:

Comments for Authors to ensure the paper complies with the Statistics Policy (Required):

Please note that the published Statistics Policy states that errors must be presented as standard deviation rather than standard error of the mean. Moreover, exact p values must be provided to three significant figures (not significant digits).

Comments to the Author:

Initial review of your manuscript, "Adenosine and acute low oxygen conditions suppress urinary bladder contractility through the activation of A2B receptors and BKCa channels" is now complete. As you will read from the attached comments, both Expert Referees find the topic and approach to carry potential impact. However, as summarized by the Reviewing Editor, both also suggest significant additional experimentation is needed before this study can achieve this potential fully. The comments of both are detailed to a sufficient extent that you can prioritize the additional work that must be performed. Please pay

particular attention to Referee 2's comments regarding the implications of your data interpretation.

In addition to the comments raised by Referee 1 pertaining to statistical powering, please review The Journal of Physiology's Statistical Policy when preparing your data for resubmission. Specifically, data variation needs to be expressed as standard deviation rather than standard error of the mean. Also, when presenting tests of significance, exact p values to three significant figures (not decimal places) are required in cases (except when $p < 0.001$).

With regard to policy on Supporting Information files, note also that data currently presented as Supplemental figures 1 and 2 need to be incorporated within figures appearing within the manuscript proper.

Please also provide the animal protocol number under the "Animals" section within the Methods.

We thank the Senior editor for their handling and thoughtful integration of the referees concerns/comments regarding our manuscript. We have addressed the concerns from both reviewers. Our response is detailed below in a point-by-point fashion. We have also conformed to the statistical presentation required by the Journal of Physiology, specifically listing p values to 3 significant figures and using standard deviation to present variability. We have also either removed or added data to the main figures regarding the supplemental data that was included in the original submission. Lastly, we provide animal protocol numbers in the Animals section of our methods.

REFeree COMMENTS

Referee #1:

This manuscript presents original work from Herrera et al., that investigates how adenosine signaling shapes normal and hypoxic urinary-bladder function. By combining an ex-vivo pressurised mouse-bladder preparation with single-cell patch-clamp recordings from detrusor smooth-muscle cells, the authors demonstrate a link between A2B adenosine-receptor activation and BK channels, which serves to dampen non-voiding pressure transients, detrusor Ca^{2+} flashes and overall bladder excitability.

The study is timely, because the field still lacks a clear mechanism linking purinergic tone, oxygen tension and detrusor contractility—a gap that is clinically relevant to overactive- and underactive-bladder syndromes as well as ischaemic bladder dysfunction.

Overall, the data add substantial new insight into the molecular mechanisms that regulate bladder excitability and offer new opportunities for novel therapeutic

strategies that target A2B receptors as treatments for bladder pathologies.

Please, find below some points that should be addressed:

-We thank the reviewer for their time and thoughtful evaluation of our manuscript. We appreciate the recognition of the current lack of mechanistic insight into how adenosine signaling, particularly under ischemic conditions, regulates bladder smooth muscle function.

The reviewer's comments and critiques have helped us improve the clarity, rigor, and overall quality of the manuscript, and we believe the revised version is strengthened as a result.

Major Revision

Reviewer Comment:

Figures 1, 2, 5, and 6 each include groups with only N = 3 experimental replicates, while Figure 4 reports N = 4. These sample sizes provide limited statistical power and insufficient biological breadth to support definitive conclusions. I therefore recommend increasing the number of independent experiments to at least $N \geq 5$ per group throughout the manuscript. If additional data has already been collected, please incorporate them; otherwise, further experiments will be required to meet this standard. Please revise the figure panels to display additional experiments/data points and update each panel's legend to reflect the updated sample sizes (revised N values).

-We appreciate the reviewer's concern regarding sample size and statistical power. In response, we have conducted additional independent experiments to increase the number of biological replicates to $N = 5$ or greater for all experimental groups in Figures 1, 2, 4, 5, and 6. The updated figures now include these additional data points, and the figure legends have been revised to reflect the new sample sizes.

Minor Revisions

1. Reviewer Comment:

Please revise the schematic in Figure 3A so that Paxilline is depicted as a BK channel inhibitor/blocker, not an activator as it is currently illustrated.

-Thank you for pointing this out. We have corrected the schematic in Figure 3A to accurately depict Paxilline as a BK channel inhibitor, not an activator. The revised figure has been updated in the manuscript.

2. Reviewer Comment:

In Panel 3E, the displayed trace suggests that Paxilline slightly lowers event frequency from control conditions, yet the summary data indicates an increase.

Please replace this trace with one that accurately represents the averaged results.

-We agree with the reviewer's observation. In response to other critiques, figure 3 has been reconfigured and now includes new data. The original paxilline treatment following adenosine administration is now removed from the manuscript. Accordingly, the new data which utilizes paxilline pretreatment clearly demonstrates the increase in transient pressure event frequency during BKCa inhibition (New Figure 3G).

3. Reviewer Comment:

In Figure 4B, the electrophysiological traces seem to show CADO alone in the second panel and Paxilline alone in the third. If the third panel actually represents the CADO + Paxilline condition, please relabel it clearly (e.g., "CADO + Paxilline") so readers can distinguish the treatments unambiguously.

-We appreciate the reviewer's attention to clarity in figure labeling. In response, the third panels in both Figure 4B and 4D are now explicitly labeled with the preceding treatments + Paxilline to avoid any ambiguity.

4. Reviewer Comment:

Lines 129-130: Please state explicitly that the phasic bladder contractions described here are cholinergic (Muscarinic) in origin.

-We thank the reviewer for this comment. In response, we have removed the data related to carbachol-induced phasic contractions from the revised manuscript. These experiments were originally included to confirm prior findings demonstrating A2B, but not A2A, receptor activity in the mouse urinary bladder, consistent with previously published work (Pakzad *et al.*, 2016; Hao *et al.*, 2019). However, as both reviewers have noted, phasic activity induced by exogenous carbachol is not equivalent to intrinsic bladder activity.

We do discuss the previous literature highlighting the lack of A2A receptor functional expression in the discussion: "In the bladder, adenosine exerts a relaxing or inhibitory effect on detrusor contractility by activating A2B receptors **and not A2A receptors** (Pakzad *et al.*, 2016; Hao *et al.*, 2019)."

Additionally, our use of the selective A2B antagonist PSB603 in the intact bladder supports the conclusion that A2B receptors mediate the observed effects on intrinsic bladder activity.

Referee #2:

This study demonstrated that the adenosine analog CADO suppresses whole bladder transient pressure rises (TPRs) and whole bladder UBSM Ca²⁺ events through the activation of A2B adenosine receptor-mediated pathway where BK channels could be involved as a downstream effector. It was also shown that acute severe hypoxic conditions attenuate both TPRs and UBSM Ca²⁺ events via A2B adenosine receptor activation, highlighting the prominent role of A2B adenosine receptor in hypoxia-associated bladder contractile disorders. I found this study important to further our understanding of mechanisms underlying ischaemic bladder, but have several concerns about interpretation of the results as well as experimental design.

-We thank the reviewer for their thoughtful and detailed evaluation of our manuscript. The reviewer's comments have helped us clarify key aspects of our experimental design and interpretation, and we believe the manuscript is significantly improved as a result. Below, we address each point in detail.

Major comments

1. Whole bladder UBSM Ca²⁺ imaging is a novel and well-designed technique that can be applied to understanding Ca²⁺ dynamics in the bladder wall. However, this technique does not seem to be effectively utilised in this study where UBSM Ca²⁺ signals are only shown as corresponding events of TPRs. Some basic information about the correlation or dissociation between the Ca²⁺ events and TPRs would be useful to demonstrate the relevance of this technique. Isn't it possible to measure the conducting velocity of the Ca²⁺ events or their synchrony amongst multiple sites? How can GCaMP signals be distinguished from contractile events, i.e., tissue distortions? Can the Ca²⁺ events be maintained in arrested bladders with wortmannin?. Showing the effects of nifedipine and paxilline alone would be helpful for the readers to realise that the Ca²⁺ events arise from LVDCC-dependent spontaneous action potentials.

-We purposefully chose the simplest and most accessible Ca²⁺ analysis in this particular study (Prevalence, Ca²⁺ activity traces, and newly added Coincidence analysis) to complement pressure results without overcomplicating the story. We feel this minimalist approach is appropriate for this study in which the main effects of treatments were not nuanced; either no effect or essentially eliminating all activity. For the reviewer's perusal, we have also included a number of Ca²⁺ analyses that were initially incorporated into this study – then removed for the sake of clarity and space.

-However, one question by referee #2 ("**Isn't it possible to measure the conducting velocity of the Ca²⁺ events or their synchrony amongst multiple sites?**") did prompt us to think about ways of how to quantify the spectrum of Ca²⁺ activity patterns observed in the detrusor - from isolated individual muscle activity, to small islands of activity, to propagating waves. We went back to the drawing board, and with the power and utility of PTCL analysis created a new form of analysis that provides valuable

information as to how Ca²⁺ activity is organized across the entire surface of the bladder. The new analysis provides novel insights into how A2B signaling and the effects of hypoxia affect the ability of Ca²⁺ events to propagate between neighboring bundles of UBSM throughout the detrusor,

Ca²⁺ events and TPRs (Intravesical Pressure):

Unfortunately we did not plan the Ca²⁺ experiments with intravesical pressure in mind, and are unable to show correlations of Ca²⁺ activity and TPRs. Specifically, we did not have pressure transducer output synchronized to camera frames. Similarly, we only imaged one hemisphere of the bladder in this study and cannot account for Ca²⁺ activity in the hidden hemisphere that may affect pressure. Our PRIM imaging setup is more effective in capturing full bladder surface Ca²⁺ activity to detail the relationship between Ca²⁺ activity and intravesical pressure generation.

Velocity Measurements:

Working with projected spherical/curved surfaces creates some difficulties in trying to analyze velocities. While we could mathematically “undistort” the projected bladder hemisphere to create a topologically correct surface (cosine transforms using surface angles between -83° to +83° are somewhat reliable), the projected wall thickness at the outer perimeter of the bladder creates a fluorescence bloom that degrades the resolvability of Ca²⁺ events, making it difficult to measure the spatial spread of intercellular Ca²⁺ waves in those regions.

The other issue with trying to calculate Ca²⁺ activity spread in the bladder is the often incoherent nature of propagating intercellular Ca²⁺ waves. Unlike gut smooth muscle that is entrained by pacemaker/slow wave signals, the nature of Ca²⁺ event propagation in the bladder depends on cell-to-cell transmission of electrical activity and paracrine transmission which can be quite variable due to a plethora of influences including: refractoriness, stretch, mucosal influences (e.g. ATP) and many more that affect the excitability of each smooth muscle cell. To illustrate this, we prepared a figure (see below) that illustrates the “wave-to-wave” variability of sequential Ca²⁺ event excursions emanating from a relatively stable initiation site at the lower right hand corner. Time was color coded according to the look up table. No two waveform patterns were the same, with variable propagation characteristics (distance between color transitions) and haphazard timing in relation to other initiation sites.

It may be the case that the area, speed and direction of propagating Ca²⁺ activity are somewhat inconsequential to overall detrusor force production and resulting pressure generation. Instead, the cumulative surface area undergoing Ca²⁺ events/contraction may be the primary determinant how much pressure is generated, and for how long. We are currently investigating this relationship.

Coincidence Analysis:

At the core of most activity patterns that occur in syncytia, is whether activity can spread, and/or be regenerated by neighboring cells. Time delays between adjacent cell activation lead to changes in velocity, direction and distance traveled which set the conditions for a range of behaviors from single isolated cell activity to synchronous activation of the entire bladder, and everything in between. Using a restricted search window approximately the size of cell and its neighbors (radius ~1mm), we measured the amount of coincident Ca²⁺ activity occurring around each active pixel of every Ca²⁺ event that occurred during the recordings (see Methods). The resulting maps and analysis show regional differences in how Ca²⁺ activity spreads in the detrusor that appear to be independent of the overall amount (frequency and duration) of Ca²⁺ activity. Although it is early days with this analysis, it offers a glimpse into a world of signaling that is normally difficult to appreciate – that is, how does conductivity of muscle activity change in real time throughout the entire bladder. We hope the updated Figures (2 & 6) go some distance in addressing the Referee's questions.

Filtering out motion artifacts:

For the large part, the distortions produced by contractions at the holding pressure used in this study (12mmHg) were minimal. Contractions were largely isometric – necessitating the use of a Ca²⁺ indicators to properly assess the extent of smooth muscle activation. In other ongoing studies in which we use pressure ramps, we see a gradual decline in tissue wall movements (displacement/distortions) at pressures

greater than 5-7mmHg. The small distortions at 12mmHg were easy to distinguish and filtered out based on particle orientation, size and width compared to bona fide Ca²⁺ events (see panel below).

Wortmannin, Nifedipine and Paxilline:

The reviewer's suggestion of examining bladder wall Ca²⁺ activity ± pressure in the presence of each of the aforementioned drugs is worthy and interesting – but is beyond the narrow and focused scope of this particular study. We are currently studying the effect of these drugs in other projects where we are better able to manipulate pressure (pressure ramping & stepping) and image the full bladder surface (PRIM) to allow for better interpretation of the effects of: 1) preventing force generation, 2) blocking voltage-gated Ca²⁺ channels and 3) blocking BK channels.

Removed Analysis:

Radial Prevalence Mapping: Calculates amount of Ca²⁺ activity in concentric rings emanating from the bladder's center – that allows some appreciation of the rhythmicity and propagation of Ca²⁺ events to be portrayed over time (ST map).

Incidence mapping. Involves combining overlapping Ca²⁺ events on multiple frames into spatio-temporal objects (STOs: 4D). The number of STOs occurring at every pixel of the bladder surface allows an appreciation of the spatial variability in the frequency of Ca²⁺ events.

2. Ln 180 (Fig.4). In mouse UBSM where iberiotoxin increases the amplitude of spontaneous action potentials and prolongs their repolarising phase, the peak value of normal action potentials is below +10mV (Hayase et al., (2009). J Urol. 181:2355-65). Does CADO significantly increase paxilline-sensitive outward currents at the physiological membrane potential range, e.g., between -50 mV and +10 mV? Ln 165 (Fig.3). Since paxilline alone should enlarge TPRs, adenosine may be capable of suppressing enlarged TPRs in paxilline-treated bladders. Therefore, effects of CADO on TPRs in bladders that had been pre-treated with paxilline should be tested.

-We thank the reviewer for raising this important point regarding the physiological relevance of BK channel activation by adenosine, particularly within the membrane potential range of -50 mV to +10 mV.

We agree that the peak of spontaneous action potentials in mouse UBSM typically does not exceed +10 mV, as reported by Hayase et al. (2009). In our voltage-clamp recordings, we did not observe a statistically significant increase in paxilline-sensitive outward currents in response to CADO at membrane potentials below +10 mV. However, this result is not entirely unexpected. We do not believe that the patch-clamp experimental design is ideally suited for assessing the impact of adenosine on BK

channels under physiological conditions. Rather, it provides a clean and efficient approach to determine whether A₂B receptor stimulation can enhance whole-cell BK channel activity. Where meaningfully observable currents under voltage clamp conditions occur at relatively extreme voltages. It is likely that slight changes in overall BK channel activity which are not detectable using whole cell voltage clamp protocol are what mediate physiological hyperpolarizing input into the bladder. Here we demonstrate that adenosine can enhance BK channel activity.

To directly address the reviewer's concern about physiological relevance, we conducted a new series of experiments using intact, pressurized bladders. In these experiments, bladders were preincubated with paxilline and then exposed to CADO. As expected, paxilline treatment alone increased pressure amplitude and frequency. Importantly, subsequent application of CADO failed to significantly reduce these pressure events. This suggests that the potent relaxing effects of CADO are mediated by BK channel activation, and that when BK channels are pharmacologically blocked, adenosine is no longer able to exert its suppressive effects on UBSM activity.

Taken together, the results from our patch-clamp recordings and the intact bladder experiments provide compelling evidence that BK channels are likely downstream targets of A₂B receptor activation in the context of physiological filling pressures. These findings have been incorporated into the revised manuscript (see Figure 3F and 3G).

Minor comments

1. Abstract Ln 49. Since the pharmacological blockade of A2B receptor with PSB 603 did not enhance TPRs (Fig.1B, C) or UBSM Ca²⁺ events (Fig.2B, C), it appears that adenosine is not continuously produced in the normal bladder to regulate physiological bladder contractility.

-Thank you for recognizing that we do not put this important fact into context in our manuscript. Accordingly, we have added the following text to our discussion:

“In our experiments, preincubation with an A2B receptor antagonist did not alter bladder contractility. This suggests that A2B receptors are not active under basal conditions. However, this does not rule out the possibility of constitutive adenosine release and A2B receptor stimulation under basal conditions in vivo. Unlike the in vivo bladder, the ex vivo bladder lacks vascular delivery of oxygen, glucose, and other substances, as well as input from efferent nerve terminals. In vivo, the bladder is subject to dynamic fluctuations in oxygen and glucose delivery and neurotransmitter input. Indeed, Hao et al. observed dramatic alterations in mouse voiding behavior when A2B receptor function was ablated (Hao et al., 2019).”

2. Ln 126 (Fig.S1). Since the TPRs arise from spontaneous UBSM contractions that are not sensitive to atropine, effects of A2A or A2B receptor agonists on

spontaneous phasic contractions rather than CCh-induced oscillatory contractions should be examined.

-This is an excellent point and was also raised by Reviewer 1 (Minor Comment 4). In response, we have removed the carbachol-induced contraction data from the manuscript. These experiments were originally intended as a higher-throughput method to confirm functional expression of A2A and A2B receptors, consistent with prior work (see response citations to reviewer 1). However, we agree that spontaneous phasic activity in the intact bladder is a more physiologically relevant context for assessing receptor function. Our revised manuscript now focuses exclusively on intrinsic bladder activity, and we highlight the role of A2B receptors using both CADO and the selective antagonist PSB603 in this setting.

3. Ln 196 (Fig.S2). Hypoxic conditions seem to be extreme. In guinea-pig bladder outlet obstruction model, the lowest oxygen saturation in the bladder wall even during voiding is about 40% (Scheepe et al. (2011), J Urol. 186:1128-33).

-We appreciate the reviewer's observation and agree that the hypoxic conditions used in our study are more severe than those typically observed in vivo. Our goal was to model acute, severe hypoxia in order to explore the signaling pathways activated under controlled hypoxic conditions. By defining a role for A2B adenosine receptors in mediating the suppression of excitation–contraction coupling in detrusor smooth muscle, we aim to lay the groundwork for future studies that examine the effects of more physiological levels of hypoxia—such as those that may occur during normal bladder function (e.g., transient reductions in blood flow during micturition).

The study by Scheepe et al. (2011), which the reviewer cited, provides a valuable reference point. In that work, the authors used spectroscopic techniques to measure bladder wall oxygen saturation, reporting values around 40% HbO₂ saturation during voiding in obstructed animals. Based on the oxyhemoglobin dissociation curve, this level of saturation corresponds to a partial pressure of oxygen (pO₂) of approximately 20–25 mmHg, which at atmospheric pressure (~760 mmHg) would equate to roughly 3% O₂ in solution.

In our study, we measured <1% O₂ in solution, which approaches anoxic conditions. We emphasize the relatively dramatic hypoxic conditions in the manuscript. As a next step, we agree that establishing an aeration gas mixture that yields ~3% O₂ in solution would be an excellent starting point for future investigations into the physiological effects of hypoxia on detrusor function.

4. Ln 196. The complete restoration of TPRs (Fig.5A, B) or Ca²⁺ events (Fig.6A, B) upon the blockade of A2B adenosine receptor in hypoxic conditions is quite surprising. Even without the inhibitory actions of adenosine, UBSM contractions

are expected to be diminished due to, for example, the severely reduced ATP production. How long were the bladders situated in the hypoxic conditions?

-We appreciate the reviewer's insightful observation regarding the restoration of TPRs (Fig. 5A, B) and Ca²⁺ events (Fig. 6A, B) under hypoxic conditions following A₂B adenosine receptor blockade. We agree that this finding is indeed surprising, especially considering the expected metabolic limitations such as reduced ATP production during hypoxia.

We believe that the primary mechanism underlying the hypoxia-induced suppression of UBSM activity is likely mediated by adenosine release within the bladder. While identifying the precise cellular source of adenosine under hypoxic conditions is beyond the scope of the current study, we speculate that the urothelium is a likely contributor, as previously suggested in the literature (Birder & Andersson, 2013; Andersson, 2015; Gutierrez Cruz *et al.*, 2022). This hypothesis aligns with prior findings that the urothelium can release ATP and adenosine in response to various stimuli, including hypoxia.

To address the reviewer's question regarding the duration of hypoxic exposure, we have clarified in the revised Methods section that bladders were typically exposed to hypoxic conditions for 15–25 minutes for each hypoxic bout.

We agree that further investigation into the source and regulation of adenosine release during hypoxia would be a valuable direction for future research. Such studies could involve targeted approaches to isolate urothelial contributions or explore the role of ectonucleotidases in adenosine metabolism under hypoxic stress.

We also appreciate the reviewer's concern regarding the potential for ATP depletion under hypoxic conditions. While prolonged or severe hypoxia can indeed lead to significant reductions in ATP levels, we believe that ATP depletion is unlikely to be a major factor during the relatively short 15–25 minute hypoxic exposures used in our experiments. This is evident during preincubation with PSB 603 where transient pressure events and Ca²⁺ are unaltered and A₂B blockade would not be expected provide an ATP-modifying role.

That said, we acknowledge that prolonged hypoxia would likely lead to ATP depletion, which could profoundly affect motor protein function and the generation of action potentials. However, within the time window of our experiments, we believe the observed suppression of UBSM activity is more plausibly attributed to adenosine-mediated signaling rather than direct energetic failure.

5. Ln 196. Effects of paxilline on the hypoxia-induced suppression of TPRs or Ca²⁺ events should be tested. Alternatively, the last sentence in Abstract and corresponding descriptions in the main text should be rewritten.

-Thank you for highlighting this potential gap in the overall signaling steps from hypoxia to adenosine receptors to decreased contractility (potentially due to BK stimulation).

We have reexamined all manuscript text have carefully avoided stating that hypoxia stimulates BK channels or that hypoxia effects are augmented by BK channel inhibition.

The data suggest that the effects adenosine are transduced by A2B receptors and downstream stimulation of BKCa channels. The observation that hypoxia effects are completely reversed by A2B receptor blockade. However, we did not explore the contribution of BK channels specifically downstream of a hypoxia challenge but would expect the same signaling cascade downstream of A2B receptors to be involved.

CITATIONS:

Andersson KE. (2015). Purinergic signalling in the urinary bladder. *Auton Neurosci* **191**, 78-81.

Birder L & Andersson KE. (2013). Urothelial signaling. *Physiol Rev* **93**, 653-680.

Gutierrez Cruz A, Aresta Branco MSL, Perrino BA, Sanders KM & Mutafova-Yambolieva VN. (2022). Urinary ATP Levels Are Controlled by Nucleotidases Released from the Urothelium in a Regulated Manner. *Metabolites* **13**.

Hao Y, Wang L, Chen H, Hill WG, Robson SC, Zeidel ML & Yu W. (2019). Targetable purinergic receptors P2Y12 and A2b antagonistically regulate bladder function. *JCI Insight* **4**.

Pakzad M, Ikeda Y, McCarthy C, Kitney DG, Jabr RI & Fry CH. (2016). Contractile effects and receptor analysis of adenosine-receptors in human detrusor muscle from stable and neuropathic bladders. *Naunyn Schmiedebergs Arch Pharmacol* **389**, 921-929.

Dear Dr Klug,

Re: JP-RP-2025-289080R1 "Adenosine and acute low oxygen conditions suppress urinary bladder contractility through the activation of A2B receptors and BKCa channels" by Gerald M. Herrera, Jason L Rengo, Grant W. Hennig, Thomas Heppner, Alexandria M Hepp, Maria Sancho, Saul Huerta de la Cruz, Mark T Nelson, and Nicholas R. Klug

Thank you for submitting your manuscript to The Journal of Physiology. It has been assessed by a Reviewing Editor and by 2 expert referees and we are pleased to tell you that it is acceptable for publication following satisfactory revision.

REVISION CHECKLIST:

Please upload two versions of your manuscript text: one with all relevant changes highlighted and one clean version with no changes tracked. The manuscript file should include all tables and figure legends, but each figure/graph should be uploaded as separate, high-resolution files. The journal is now integrated with Wiley's Image Checking service. For further details, see: <https://www.wiley.com/en-us/network/publishing/research-publishing/trending-stories/upholding-image-integrity-wileys->

image-screening-service

We look forward to receiving your revised submission.

Yours sincerely,

Peying Fong
Senior Editor
The Journal of Physiology

EDITOR COMMENTS

Reviewing Editor:

Thank you for your revised manuscript. There are some minor suggestions from reviewer 2 to improve the readability of the text that we suggest incorporating in a final minor revision.

Senior Editor:

Review of your revised manuscript, "Adenosine and acute low oxygen conditions suppress urinary bladder contractility through the activation of A2B receptors and BKCa channels" is now complete. Thank you for being responsive to feedback offered in the initial review cycle.

I commend you to the referee reports (below), and the brief summary from the Reviewing Editor (above).

At this time, there are three suggestions for improvement offered by Referee 2 that I anticipate you can address readily.

We look forward to receiving your revised manuscript and thank you for submitting your work for consideration by The Journal of Physiology.

REFeree COMMENTS

Referee #1:

The reviewer finds that the authors have satisfactorily addressed all of my comments and therefore recommends the manuscript for publication.

Referee #2:

The authors have adequately addressed all of my concerns. The additional Ca²⁺ event measurement, namely 'coincidence' and 'spatio-temporal maps' demonstrating the pattering of Ca²⁺ activity (Fig 2 and 6) provides a clearer understanding of Ca²⁺ dynamics in ex vivo whole bladders.

There are few minor comments that need to be addressed.

Ln 197-198. 'Adenosine signaling via A2B receptor activation promotes downstream cAMP production and protein kinase A (PKA) activation'. This should be introduced earlier. For example, in Ln 149 to clarify why the effects of forskolin on transient pressure events was examined.

Ln 349-350. 'Indeed, adenosine potently increased paxilline-sensitive outward currents. An effect that was sensitive to pre-treatment with the A2B receptor antagonist PSB 603 (Fig. 4 B, D)'. For the readers who are familiar with electrical properties of detrusor smooth muscles, it would be helpful to state, for example : 'slight increases in BK activity within physiological membrane potential range (between -50 and +10 mV) would be sufficient to suppress action potential-triggered detrusor contractility.'

Ln 381. Since ATP production that is critical for detrusor muscle contractions is expected to decrease during hypoxia, it would be better to state, for example: 'ATP appears not to be depleted during relatively short (15-25 min) severe chemical hypoxia'.

END OF COMMENTS

The authors have adequately addressed all of my concerns. The additional Ca²⁺ event measurement, namely 'coincidence' and 'spatio-temporal maps' demonstrating the pattering of Ca²⁺ activity (Fig 2 and 6) provides a clearer understanding of Ca²⁺ dynamics in ex vivo whole bladders.

There are few minor comments that need to be addressed.

Ln 197-198. '*Adenosine signaling via A2B receptor activation promotes downstream cAMP production and protein kinase A (PKA) activation*'. This should be introduced earlier. For example, in Ln 149 to clarify why the effects of forskolin on transient pressure events was examined.

Ln 349-350. '*Indeed, adenosine potently increased paxilline-sensitive outward currents. An effect that was sensitive to pre-treatment with the A2B receptor antagonist PSB 603 (Fig. 4 B, D)*'. For the readers who are familiar with electrical properties of detrusor smooth muscles, it would be helpful to state, for example : 'slight increases in BK activity within physiological membrane potential range (between -50 and +10 mV) would be sufficient to suppress action potential-triggered detrusor contractility.'

Ln 381. Since ATP production that is critical for detrusor muscle contractions is expected to decrease during hypoxia, it would be better to state, for example: 'ATP appears not to be depleted during relatively short (15-25 min) sever chemical hypoxia'.

Herrera et al., response to editors and reviewers.

Editor and Reviewer 2 comments in **blue** and responses in **black**. Responses incorporated into the manuscript are in **red**.

EDITOR COMMENTS

Reviewing Editor:

Thank you for your revised manuscript. There are some minor suggestions from reviewer 2 to improve the readability of the text that we suggest incorporating in a final minor revision.

Thank you for handling the manuscript as reviewing editor. We have incorporated text changes suggested by reviewer 2. Exact changes detailed below.

Senior Editor:

Review of your revised manuscript, "Adenosine and acute low oxygen conditions suppress urinary bladder contractility through the activation of A2B receptors and BKCa channels" is now complete. Thank you for being responsive to feedback offered in the initial review cycle.

I commend you to the referee reports (below), and the brief summary from the Reviewing Editor (above).

At this time, there are three suggestions for improvement offered by Referee 2 that I anticipate you can address readily.

We look forward to receiving your revised manuscript and thank you for submitting your work for consideration by The Journal of Physiology.

We appreciate your time and effort as senior editor for our manuscript. We have made changes/improvement as suggested by reviewer 2 below.

REFeree COMMENTS

Referee #1:

The reviewer finds that the authors have satisfactorily addressed all of my comments and therefore recommends the manuscript for publication.

We thank reviewer 1 for their time and quality assessment and critique of the manuscript.

Referee #2:

The authors have adequately addressed all of my concerns. The additional Ca²⁺ event measurement, namely 'coincidence' and 'spatio-temporal maps' demonstrating the pattering of Ca²⁺ activity (Fig 2 and 6) provides a clearer understanding of Ca²⁺ dynamics in ex vivo whole bladders.

We thank reviewer 2 for their detailed comments and critiques. We further thank reviewer 2 for their initial comments regarding Ca²⁺ signaling, thus provoking the improved coincident analysis.

There are few minor comments that need to be addressed.

Ln 197-198. 'Adenosine signaling via A2B receptor activation promotes downstream cAMP production and protein kinase A (PKA) activation'. This should be introduced earlier. For example, in Ln 149 to clarify why the effects of forskolin

on transient pressure events was examined.

This is a great suggestion. We have now deleted the sentence at Ln 197 and added this wording to the suggested text at Ln 149:

“This relaxing effect was abolished by preincubation with A2B receptor antagonist PSB 603 (500 nM, Fig. 1B), suggesting that the relaxing effects of adenosine occur through A2B receptor-mediated Gs-protein coupled receptor (GsPCR) **stimulation and downstream cAMP to protein kinase A (PKA) signaling**. Indeed, forskolin (5 μ M), an adenylate cyclase activator, mimicked the relaxing effects of CADO even in the presence of A2B receptor blockade (Fig. 1B), confirming that the mechanism is cyclic adenosine monophosphate (cAMP)-dependent. Representative traces of transient pressure events for the different experimental conditions are shown in Figure 1C.”

Ln 349-350. 'Indeed, adenosine potently increased paxilline-sensitive outward currents. An effect that was sensitive to pre-treatment with the A2B receptor antagonist PSB 603 (Fig. 4 B, D)'. For the readers who are familiar with electrical properties of detrusor smooth muscles, it would be helpful to state, for example : 'slight increases in BK activity within physiological membrane potential range (between -50 and +10 mV) would be sufficient to suppress action potential-triggered detrusor contractility.'

Thank you, we have now included the following wording into this part of the discussion:

“Bladder smooth muscle K_{ATP} channels showed no apparent sensitivity to adenosine. However, PKA can also increase activity of smooth muscle BK_{Ca} channels. The UBSM BK_{Ca} channel serves a critical role in regulating bladder excitability due to its outsized contribution to UBSM action potential repolarization. Notably, even slight increases in BK_{Ca} channel activity within the UBSM physiological membrane potential range of -50 to +10 mV are sufficient to suppress smooth muscle contractility (Schubert & Nelson, 2001; Sancho & Kyle, 2021). Based on this, we investigated whether A2B receptor stimulation enhances BK_{Ca} channel activity in UBSM. Since K_{ATP} channels had no apparent sensitivity to adenosine and smooth muscle BK_{Ca} channels are also strongly modulated by PKA activity, we investigated whether UBSM BK_{Ca} activity was enhanced following A2B receptor stimulation.”

Ln 381. Since ATP production that is critical for detrusor muscle contractions is expected to decrease during hypoxia, it would be better to state, for example: 'ATP appears not to be depleted during relatively short (15-25 min) severe chemical hypoxia'.

Thank you for this comment. It is important that we provide clear language about our timing and potential effects of longer duration hypoxia. Accordingly, we have added this language to that part of the discussion:

“While our model reflects acute and severe hypoxia, we reasoned that it offers a controlled platform to investigate fundamental signaling mechanisms in a

reproducible and robust manner. **The 15–25 minute acute hypoxia treatment does not appear to directly impair ATP-sensitive processes. However, prolonged exposure to low oxygen conditions would be expected to eventually reduce cytosolic ATP levels, leading to dysfunction in ATP-dependent mechanisms such as contractile protein function.** Our findings indicate that A2B receptor stimulation is critical for the relaxing effects of severe chemical hypoxia, supporting a direct link between hypoxia and adenosine signaling in the urinary bladder. “

Dear Dr Klug,

Re: JP-RP-2025-289080R2 "Adenosine and acute low oxygen conditions suppress urinary bladder contractility through the activation of A2B receptors and BKCa channels" by Gerald M. Herrera, Jason L Rengo, Grant W. Hennig, Thomas Heppner, Alexandria M Hepp, Maria Sancho, Saul Huerta de la Cruz, Mark T Nelson, and Nicholas R. Klug

We are pleased to tell you that your paper has been accepted for publication in The Journal of Physiology.

Yours sincerely,

Peying Fong
Senior Editor
The Journal of Physiology

If you would like to receive our 'Research Roundup', a monthly newsletter highlighting the cutting-edge research published in The Physiological Society's family of journals (The Journal of Physiology, Experimental Physiology, Physiological Reports, The Journal of Nutritional Physiology and The Journal of Precision Medicine: Health and Disease), please click this link, fill in your name and email address and select 'Research Roundup':
<https://www.physoc.org/journals-and-media/membernews>

- **TRANSPARENT PEER REVIEW POLICY:** To improve the transparency of its peer review process, The Journal of Physiology publishes online as supporting information the peer review history of all articles accepted for publication. Readers will have access to decision letters, including Editors' comments and referee reports, for each version of the manuscript as well as any author responses to peer review comments. Referees can decide whether or not they wish to be named on the peer review history document.
- You can help your research get the attention it deserves! Check out Wiley's free Promotion Guide for best-practice recommendations for promoting your work at: www.wileyauthors.com/eeo/guide. You can learn more about Wiley Editing Services which offers professional video, design, and writing services to create shareable video abstracts, infographics, conference posters, lay summaries, and research news stories for your research at: www.wileyauthors.com/eeo/promotion.
- **IMPORTANT NOTICE ABOUT OPEN ACCESS:** To assist authors whose funding agencies mandate public access to published research findings sooner than 12 months after publication, The Journal of Physiology allows authors to pay an Open Access (OA) fee to have their papers made freely available immediately on publication.

EDITOR COMMENTS

Reviewing Editor:

Thank you for revising your manuscript and submitting such a high quality study to the Journal.

Senior Editor:

Thank you for thoroughly addressing all residual points, and especially for contributing this fine study to The Journal of Physiology. Congratulations!